# FAST GENERIC INTERACTION DETECTION FOR MODEL INTERPRETABILITY AND COMPRESSION

**Tianjian Zhang**[1,2]**, Feng Yin**[1,2]**, Zhi-Quan Luo**[1,2]
[1]School of Science and Engineering, The Chinese University of Hong Kong, Shenzhen
[2]Shenzhen Research Institute of Big Data
`tianjianzhang@link.cuhk.edu.cn` `{yinfeng,luozq}@cuhk.edu.cn`

## ABSTRACT

The ability of discovering feature interactions in a black-box model is vital to explainable deep learning. We propose a principled, global interaction detection method by casting our target as a multi-arm bandits problem and solving it swiftly with the UCB algorithm. This adaptive method is free of ad-hoc assumptions and among the cutting-edge methods with outstanding detection accuracy and stability. Based on the detection outcome, a lightweight and interpretable deep learning model (called ParaACE) is further built using the alternating conditional expectation (ACE) method. Our proposed ParaACE improves the prediction performance by 26% and reduces the model size by 100+ times as compared to its Teacher model over various datasets. Furthermore, we show the great potential of our method for scientific discovery through interpreting various real datasets in the economics and smart medicine sectors. The code is available at https://github.com/zhangtj1996/ParaACE.

## 1 INTRODUCTION

Explainable machine learning is an active research field that focuses on providing interpretable models, transparent explanations, and confident decisions to practical AI systems. Investigating feature interaction is vital to model interpretability. Interaction detection should be able to reveal which subset of features influence the output *jointly*, and what the corresponding nonlinear transformation is. We aim to find the underlying interactions from data, such that we can interpret the model properly. To go one step further, we hope that new models can be built with the aid of the detected interaction knowledge economically to avoid heavy parameterization.

In this paper, we first restrict ourselves to interaction detection. A novel method is derived directly from the most acknowledged definition of feature interaction (Friedman et al., 2008). We further apply the obtained interaction knowledge to design a transparent and refined neural network (NN). This approach fits well into the data science life cycle (Yu & Kumbier, 2019), which was applied, for instance in Tsang et al. (2020a), successfully for interpretable recommender system design.

The main contributions of this paper include: 1) a fast and principled interaction detection method, 2) a lightweight and interpretable neural network model that can surpass its Teacher, 3) thorough theoretical analysis and performance evaluations with real datasets. More details are given below.

1. We propose a generic interaction detection method based on a global statistical metric, namely the expected Hessian, $H_{ij} := E_{\mathbf{x}} \left[ \frac{\partial^2 F(\mathbf{x})}{\partial x_i \partial x_j} \right]$. Notably, our method is **model-agnostic** and applicable to any pre-trained learning model, $F(\mathbf{x})$, being for instance, a deep neural network model or a tree model. It is also flexible to use for multi-way interaction detection.

2. To speed up the detection process, we evaluate the expected Hessian via adaptive sampling using the Upper Confidence Bound (UCB) algorithm (Lai & Robbins, 1985), which can significantly reduce the computational complexity. Besides, thorough analysis of the proposed interaction detection method are conducted.

3. Using the detected interaction pairs, we further design a compressed but interpretable Student model which can surpass its Teacher by **26%** in terms of data fitting performance

averaged over various datasets. The compressed model reduces its size over **100 times** compared to the baseline fully-connected, over-parameterized neural network (OverparaFC).

4. We demonstrate the linkages between our compressed model and the classic alternating conditional expectation (ACE) model (Breiman & Friedman, 1985).

5. We conduct large-scale performance evaluations and further explain the obtained model interpretability with some real datasets.

The remainder of this paper is organized as follows. Section 2 introduces all related works. Our proposed interaction detection method is introduced in Section 3. By exploiting the detection outcome, a new variant of lightweight and interpretable deep learning model is introduced in Section 4. Experimental results are given in Section 5. Finally, we conclude the paper in Section 6.

## 2 RELATED WORKS

**Interaction Detection:** Early works adopt pure statistics for detecting feature interactions, and representatives include ANOVA and GUIDE (Wonnacott & Wonnacott, 1990; Fisher, 1992; Loh, 2002). These works have motivated a plethora of new methods with the aim to enhance the detection accuracy and/or efficiency. The first class of methods centered around the $GA^2M$ and tree models, see for instance Lou et al. (2013); Sorokina et al. (2008); Friedman et al. (2008); Lundberg et al. (2020). The second class of methods were built on the so-called factorization machines (Rendle, 2010) as well as its new variants (Xiao et al., 2017; Song et al., 2019) with attention mechanism. The third class exploits the most recent advances in deep learning, including the Neural Interaction Detection (NID) and some new variants (Tsang et al., 2018a;b; Cui et al., 2019): Persistence Interaction Detection (PID) (Liu et al., 2020), Integrated Hessians (IH) (Janizek et al., 2020), Shapley interaction (Zhang et al., 2020; Sundararajan et al., 2020), etc. Although we have witnessed well improved interaction detection performance for many datasets over the decades, the above methods still lead to inconsistent detection results for some other datasets. The reasons are twofold. Firstly, the interaction strength is empirically defined, for instance, NID method computes the interaction strength via summarizing the neural network weights. Secondly, a specific deep learning model is required, for instance, an $\ell_1$-regularized ReLU network is required by the latest NID and PID methods to maintain high accuracy. In contrast, our proposed method is derived directly from the definition of feature interaction (Friedman et al., 2008) and moreover is not confined to any specific learning model.

**Model Interpretability:** There are two categories of approaches to address model interpretability, namely the transparency-based and post-hoc approaches (Došilović et al., 2018). *Transparency-based approaches* require the model itself to be simple and interpretable, like linear models, decision trees, etc. One can directly read off the interpretations from their coefficients or decision rules. But often, they are less accurate due to limited representation power. In contrast, *post-hoc approaches* extract useful information from a pre-trained model, which is often complex and hard to interpret. Well-known methods such as LIME (Ribeiro et al., 2016) and SHAP (Lundberg & Lee, 2017) fall in this category, but they did not take feature interaction into account. Recently proposed symbolic metamodel (Alaa & van der Schaar, 2019) captures nonlinear interactions by approximating the black-box model with explicit symbolic expressions. There are also some other post-hoc approaches based on sensitivity analysis (Cortez & Embrechts, 2013), which return a quantification of feature importance $\nu_i = E[(\frac{\partial F(\mathbf{x})}{\partial x_i})^2]$ (Kucherenko et al., 2009) and interactions $\nu_{ij} = E[|\frac{\partial^2 F(\mathbf{x})}{\partial x_i \partial x_j}|^2]$ (Roustant et al., 2014) by input perturbation. Our work aims to combine the strengths of the two categories. Concretely, we first extract the interaction knowledge by a post-hoc method, and then build a transparent and interpretable learning model as illustrated in Figure 3.

**Model Compression and Knowledge Distillation (KD):** Model compression (Bucilua et al., 2006) aims to learn a small compressed model (Student) from a large complex model (Teacher) with augmented training data produced by the Teacher. Compared to the Teacher, the Student can make similar or even better predictions. Knowledge distillation (Hinton et al., 2015) mainly deals with multi-class classification problems and extracts "valuable information that defines a rich similarity structure over the data". Our work introduces a novel viewpoint of knowledge (the interacted relationships) and targets a lightweight but more accurate Student model.

## 3 PROPOSED INTERACTION DETECTION METHOD

Before diving into in-depth discussions of interaction detection, we need to formally define what feature interaction is. The textbook definition according to Friedman et al. (2008) is given below.

**Definition 3.1** (Friedman & Popescu 2008). *A function $F : \mathbb{R}^p \to \mathbb{R}$ is said to exhibit an interaction between two of its variables $x_i$ and $x_j$ if the difference in the value of $F(\mathbf{x})$ as a result of changing the value of $x_i$ depends on the value of $x_j$.*

Equivalently, if $E_\mathbf{x}\left[|\frac{\partial^2 F(\mathbf{x})}{\partial x_i \partial x_j}|^2\right] > 0$, namely the partial derivative w.r.t $x_j$ turns out to be dependent on $x_i$, then we say $x_i$ and $x_j$ are interacted. Otherwise, $x_i$ and $x_j$ have no interaction, if $F(\mathbf{x})$ can be expressed as the sum of two functions $f_{\backslash i}$ and $f_{\backslash j}$ (Sorokina et al., 2008), namely,

$$F(\mathbf{x}) = f_{\backslash i}\left(x_1, \ldots, x_{i-1}, x_{i+1}, \ldots, x_p\right) + f_{\backslash j}\left(x_1, \ldots, x_{j-1}, x_{j+1}, \ldots, x_p\right),$$

where $f_{\backslash i(j)}$ is irrespective to $x_{i(j)}$. Similarly, higher-order (multi-way) interaction can be defined. In this paper, we mainly focus on pairwise interaction, while multi-way interaction is only briefly discussed due to space limitations.

### 3.1 INTERACTION STRENGTH MEASURE

Motivated by the above definition of interaction, it is natural to take advantage of the Hessian matrix $H := \nabla^2_{\mathbf{xx}} F(\mathbf{x})$. The magnitude of its entry $|\frac{\partial^2 F(\mathbf{x})}{\partial x_i \partial x_j}|$ contains rich information about the local curvature at a data point $\mathbf{x}$. Note that $F(\mathbf{x})$ is a regression function here, not a loss function. This idea was originally considered in the economics community (Ai & Norton, 2003) and rediscovered for sensitivity analysis in Roustant et al. (2014). The goodness of $F(\mathbf{x})$ as an approximator can essentially influence the interaction detection performance, see our Theorem M.2 in the supplement.

We define $g(\mathbf{x}, i, j) := |\frac{\partial^2 F(\mathbf{x})}{\partial x_i \partial x_j}|^2$ to measure the **local interaction strength** of the $i$-th and $j$-th features at point $\mathbf{x}$. We then use $f(i, j) := E_\mathbf{x}[g(\mathbf{x}, i, j)] = E_\mathbf{x}\left[|\frac{\partial^2 F(\mathbf{x})}{\partial x_i \partial x_j}|^2\right]$ as a measure of the **global interaction strength**. If $f(i, j) \approx 0$, we say feature $x_i$ and $x_j$ have weak interaction; otherwise, if $f(i, j)$ is significantly larger than zero, then $x_i$ and $x_j$ have strong interaction. **In this paper, we focus on the global interaction.**

### 3.2 INTERACTION STRENGTH EVALUATION

The above defined global interaction strength $f(i, j) = E_\mathbf{x}\left[|\frac{\partial^2 F(\mathbf{x})}{\partial x_i \partial x_j}|^2\right]$ is mostly unavailable due to the unknown input distribution $\mathbf{x} \sim P(\mathbf{x})$. The Monte Carlo method can be used to approximate it by computing the sample mean over the training data(Roustant et al., 2014).

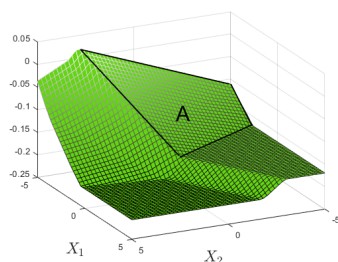

Figure 1: The landscape of a ReLU network with two inputs $(X_1, X_2)$.

**Analytical Evaluation:** The Hessian matrix $\nabla^2_{\mathbf{xx}} F(\mathbf{x})$ for neural networks at a certain data point can be calculated analytically and efficiently, by using the automatic differentiation (Paszke et al., 2017). However, using this analytical solution is problematic for some learning models, such as the ReLU network, Random Forest (RF), etc., see supplement A. For example, the landscape of a ReLU network is piece-wise linear as shown in Figure 1, thus the exact Hessian is a zero matrix at almost every point. So, we turn to the following numerical evaluation for broader horizons.

**Numerical Evaluation:** Finite difference method is a common way to approximate the Hessian on a given data point (Campolongo & Braddock, 1999), i.e.,

$$\frac{\partial^2 F(\mathbf{x})}{\partial x_i \partial x_j} \approx \frac{1}{4h_i h_j}[F(\mathbf{x} + \mathbf{e}_i h_i + \mathbf{e}_j h_j) - F(\mathbf{x} + \mathbf{e}_i h_i - \mathbf{e}_j h_j) \\ - F(\mathbf{x} + \mathbf{e}_j h_j - \mathbf{e}_i h_i) + F(\mathbf{x} - \mathbf{e}_i h_i - \mathbf{e}_j h_j)], \tag{1}$$

where $\mathbf{e}_i$ is a one-hot vector with the $i$-th element being equal to one and the rest of elements being zeros. We also note that if the computation of gradient $\frac{\partial F}{\partial \mathbf{x}}$ is cheap (e.g., $F$ is a neural network), then $H_{ij}$ can be approximated as $\frac{1}{2h}\left[\frac{\partial F(\mathbf{x}+\mathbf{e}_j h)}{\partial x_i} - \frac{\partial F(\mathbf{x}-\mathbf{e}_j h)}{\partial x_i}\right]$ to reduce computation, which is similar to the feature interaction score defined in Greenside et al. (2018). The choice of perturbation size $h_i$ or $h_j$ (abbr. $h_{i(j)}$) is critical. Generally, we do not want $h_{i(j)}$ to be too small (incurring round-off error) or too large (incurring truncation error) to get a good overall approximation of the derivative (Jerrell, 1997; Baydin et al., 2018). For our problem, we particularly do not want $h_{i(j)}$ to be too small so that the four evaluated points (shown in the numerator of Equation 1) lie on the same hyperplane (e.g., region A in Figure 1), which makes the quantity in Equation 1 always zero. The following theorem reveals that choosing a sufficiently small $h_{i(j)}$ is not necessary.

**Theorem 3.1.** *For any $x$ and $y$, function $F$ shows no interaction between them, i.e., it can be decomposed as $F(x, y) = a(x) + b(y)$ **if and only if**, **for any** $h, k > 0$, $F(x + h, y + k) - F(x + h, y - k) - F(x - h, y + k) + F(x - h, y - k) = 0$.*

The magnitude of the numerator in Equation 1 tells us whether $F$ is locally separable for variables $x_i$ and $x_j$ at point $\mathbf{x}$, see our proof in supplement B.

The main issue of the finite difference method lies in the computational complexity for approximating the global interaction strength $f(i, j)$, especially when the function evaluation itself is expensive. Using all the training samples for finding just a few strongest interaction pairs can be a total waste of computation resources. In general, for a dataset with $N$ samples in $p$-dimensional feature spaces, a total number of $4Np(p-1)/2$ arithmetic evaluations of the surrogate regression function $F$ are needed. If the Hessian happens to be sparse, that is, there are only a few interactions existing in the ground truth function, massive evaluations on those feature pairs with zero interaction strength should be avoided.

### 3.3 IDENTIFICATION OF THE $k$-STRONGEST PAIRWISE INTERACTIONS

In practice, we do not have to obtain the interaction strengths for all interaction pairs, because many of them are simply too weak to have an impact on the output, and the top $k$ strongest pairwise interactions are well sufficient for data modeling and prediction. Finding the top $k$ strongest pairwise interactions fits perfectly into the best $k$-arms identification problem in the context of multi-armed bandits.

Inspired by the recent work (Bagaria et al., 2018b), we can treat each entry of the Hessian $H_{ij}$ as an arm $\mathcal{A}_r$ (see Figure 2). Since Hessian is a symmetric matrix, we only need to consider $n = p(p-1)/2$ arms in the set $\{\mathcal{A}_r : r \in [n]\}$. For example, if we choose to pull the arm for the $i$-th row and $j$-th column of the Hessian, we will evaluate the local interaction strength $g(\mathbf{x}_\xi, i, j)$ on a randomly selected training data point $\mathbf{x}_\xi$. The strength $g(\mathbf{x}_\xi, i, j)$ can be regarded as a random reward, where $\mathbf{x}_\xi$ is uniformly drawn from the training data $\{\mathbf{x}_1, \ldots, \mathbf{x}_N\}$. For ease of notation, we let $f(r; \xi) := g(\mathbf{x}_\xi, i, j)$ as the random reward of the arm $r$, and define $\mu_r := \mathbb{E}_\xi[f(r; \xi)]$ as the true mean value. **Our goal is to find the best $k$ arms with the highest mean reward.**

This reformulation is reasonable since the following three essential assumptions for multi-armed bandits (Slivkins et al., 2019) are satisfied.

  a. Only the reward will be observed after each pull;

  b. The reward for each arm is drawn independently and identically from its reward distribution;

  c. The reward for each round is bounded. Since we only consider functions $F(\mathbf{x})$ defined on a compact set, if $F$ is a continuous function and twice differentiable, its first- and second-order derivatives are bounded accordingly.

### 3.4 DETECTING THE $k$-STRONGEST PAIRWISE INTERACTIONS WITH UCB ALGORITHM

In this section, we briefly introduce the UCB algorithm (Lai & Robbins, 1985) for finding the $k$-strongest interactions. Here, we design a sequence of estimators $\{\hat{f}_\ell(r)\}$ for the mean reward of the arm $\mathcal{A}_r$ and construct confidence interval $C(\ell)$. We let $\hat{f}_\ell(r)$ denote the estimator after $\ell$ pulls of the arm $\mathcal{A}_r$. Several assumptions need to be made before we give out the primary outcome.

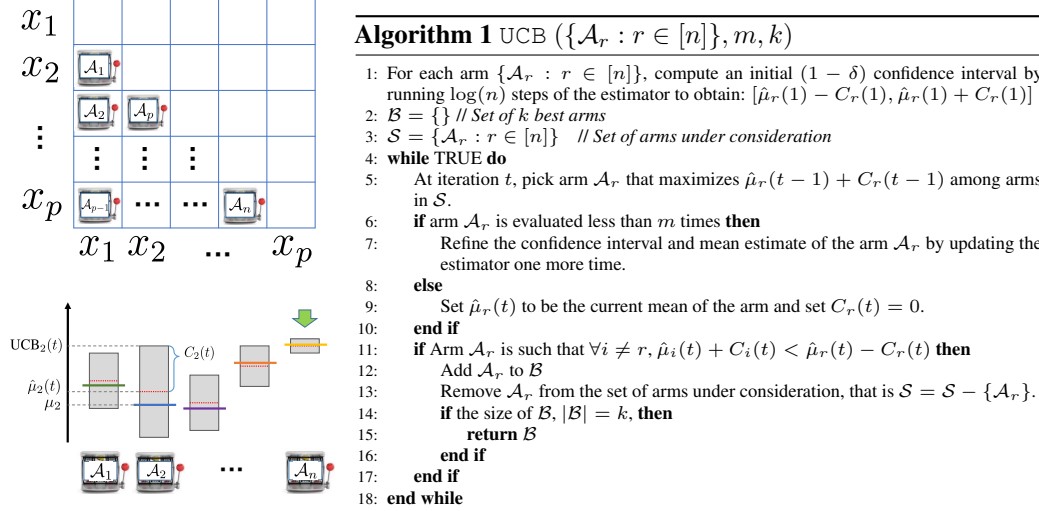

Figure 2: **Top Left:** The entries of the Hessian are treated as arms. **Bottom Left:** Illustration of the notations used in Algorithm 1. This is a snapshot taken at iteration $t$, where the colored solid bar and the red dashed bar are the true mean reward and the current estimate for each arm, respectively. At each iteration, the arm with the highest UCB will be pulled. **Right:** Pseudo-code for the UCB algorithm.

**Assumption 1.** *Finite $m$ evaluations of one arm are sufficient to obtain an accurate reward.*

**Assumption 2.** *Estimators $\hat{f}_\ell(r)$ are $\sigma_r$-subgaussian.*

Assumption 2 is satisfied because the reward is bounded (Hoeffding's lemma, 1963). Here, $\sigma_r$s are pre-defined parameters. In the following analysis, we will choose $\sigma = \max_r\{\sigma_r\}$, such that all $\hat{f}_\ell(r)$ are $\sigma$-subgaussian.

Let $\hat{f}_\ell(r) := \frac{1}{\ell}\sum_{i=1}^{\ell} f(r; \xi_i)$, and we construct the $1 - \delta$ confidence intervals (see supplement C) of $\hat{f}_\ell(r)$ as,

$$C(\ell) := \begin{cases} \sqrt{\frac{2\sigma^2 \log \frac{2}{\delta}}{\ell}} & \text{if } \ell \leq m \\ 0 & \text{if } \ell > m \end{cases}, \tag{2}$$

$$\mu_r \in \left[\hat{f}_\ell(r) - C(\ell), \hat{f}_\ell(r) + C(\ell)\right], \text{ w.p. } 1 - \delta, \tag{3}$$

where $\mu_r$ denotes the true expectation. Increasing number of pulls of one arm, the uncertainty of the reward for that arm will decrease. When the number of pulls reaches $m$, we can remove the uncertainty of the arm and set $C(\ell) = 0$. Let $\ell_r(t)$ count the number of pulls of arm $\mathcal{A}_r$ till iteration $t$. To simplify the notations, we let $\hat{\mu}_r(t) = \hat{f}_{\ell_r(t)}(r)$ and $C_r(t) = C(\ell_r(t))$ be the sample means and confidence intervals at iteration $t$. Now, we formally introduce the complete procedure in Algorithm 1.

The Algorithm 1 is initialized by evaluating all arms $\mathcal{O}(\log(n))$ times, then at each subsequent iteration, it picks the arm with the highest UCB, i.e., $\max_j\{\hat{\mu}_j(t) + C_j(t)\}$, to evaluate (see Figure 2). If one arm has been evaluated for $m$ times, we will its true mean value and set the uncertainty to zero. At the end of each iteration, if there exists one arm whose Lower Confidence Bound (LCB) is higher than all the other's UCB, we will put it into the set of $k$ best arms.

In the following, we give a theoretical analysis on the complexity of this algorithm. We have to further define $\Delta_i^{(k)} := \max(0, \mu_{i_k^*} - \mu_i)$, where $i_k^*$ is the index for the $k$-th best arm and $\Delta_i^{(k)}$ indicates the gap of true values between the top-$k$ arms and the rest.

**Theorem 3.2.** *With probability $1 - \Theta(\frac{1}{n^2})$, given maximal pulling times $m$ for each arm, Algorithm 1 returns the $k$-strongest interaction pairs in $\mathcal{O}\left(\sum_{i=1}^{n} \log(n)\left(\frac{\sigma^2 \log(nm)}{(\Delta_i^{(k)})^2} \wedge m\right)\right)$ time, where $(\cdot \wedge \cdot)$ is short for $\min(\cdot, \cdot)$.*

**Remark 3.1.** *When $k \ll n$, the complexity of Algorithm 1 for finding the top $k$ interactions is $\mathcal{O}(n \log(mn) \log(n) + km \log(n))$ under some natural assumptions on the distribution of $\Delta_i$, being superior to the naive algorithm (every arm is pulled $m$ times) with complexity $\mathcal{O}(nm)$.*

**Remark 3.2.** *For $k = 1$, namely the strongest interaction pair identification, it requires at most $\sum_{i=1}^{n} ((\frac{8\sigma^2}{(\Delta_i^{(1)})^2} \log(n^3 m)) \wedge m)$ pulls. We leave the complete proof in supplement C.*

### 3.5 EXTENSION TO MULTI-WAY INTERACTION

Similarly, the third-order derivatives $\frac{\partial^3 F}{\partial x_i \partial x_j \partial x_k}, \forall (i, j, k)$, and even the higher-order ones can be approximated by the finite difference method as well, but the computational complexity will increase geometrically. For a remedy, one can detect higher-order interactions from the obtained pairwise ones. For instance, based on the $k$-strongest pairwise interaction strengths, we can further construct an undirected graph $\mathcal{G}$, in which each node represents a feature. We develop Algorithm 2 to detect the multi-way interactions through checking all the cliques in the graph $\mathcal{G}$, and more details are given in supplement G. Despite the computation complexity, our proposed algorithm is more advantageous to use because the associated learning model is agnostic and the form of interaction is adaptive as compared with some existing methods, such as Min et al. (2014), which relies on a logistic regression model and multiplicative interactions of a particular form.

## 4 MODEL COMPRESSION WITH DETECTED INTERACTIONS

The detected interactions can be used for designing structured and more transparent models. A similar idea was mentioned in Tsang et al. (2020b). But we propose a new architecture called Parametric ACE (ParaACE), inspired by several seminal works on Generalized Additive Model (GAM) (Hastie, 2017) and (nonparametric) Alternating Conditional Expectation (ACE) method (Breiman & Friedman, 1985). The original ACE method is based on the following model,

$$\theta(Y) = \sum_{k=1}^{p} \phi_k(X_k) + \epsilon, \tag{4}$$

and solves for the optimal transformation functions $\{\theta, \phi_1, \cdots, \phi_p\}$ nonparametrically in the Hilbert space. However, it only considers transformation of each single feature and is time consuming for large feature dimension $p$. We build the ParaACE model with the detected interactions as

$$h(\mathbf{x}; \{\boldsymbol{\theta}_{s_i}\}, \{\boldsymbol{\theta}_{r_i}\}, \boldsymbol{\beta}) := \beta_0 + \sum_{i=1}^{p} \beta_i s_i(x_i; \boldsymbol{\theta}_{s_i}) + \sum_{i=1}^{R} \beta_{p+i} r_i(\mathbf{x}_{\mathcal{I}_i}; \boldsymbol{\theta}_{r_i}), \tag{5}$$

where $\{\mathcal{I}_i\}_{i=1}^{R}$ is a set of indices for the detected interaction pairs, $s_i$ and $r_i$ are the transformation functions for a single feature and interacting features with the corresponding parameters $\{\boldsymbol{\theta}_{s_i}\}$ and $\{\boldsymbol{\theta}_{r_i}\}$, $\boldsymbol{\beta}$ is a weight vector for the output of all the transformation functions.

The new architecture is shown in Figure 3 with two meta layers: the *optimal feature transformation* layer (Equation 5, boxed in red and green) and an optional *fix-up* layer (boxed in yellow). The optimal transformation layer gives the transformation functions of both the main effects and interacting features explicitly. The fix-up layer can be understood as the inverse function for the optimal transformation of the output, $\theta(Y)$, in the original ACE (Equation 4). Another functionality of the fix-up layer is to alleviate the negative impacts of the wrongly detected interactions on the output. More explanations are given in supplement H.

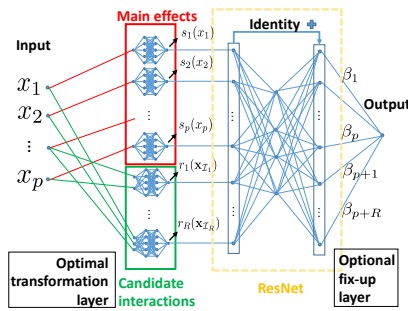

Figure 3: Parametric ACE model.

We highlight the advantages of this model as follows: 1) **interpretable**, unlike many other models, ParaACE model gives explicit transformations of features, making it interpretable. 2) **lightweight**, this model can significantly reduce the number of parameters from the original one, and unlike the pruning technique (Han et al., 2015), it does not need extra memory to store the indices of the preserved weights.

3) **flexible**, the structure of NN is modularized by blocks, which is easy to plug in/off for other applications. The fix-up layer can be redesigned as block-wise subnetworks to capture high-level interactions, end up in the shape of a hierarchical deep network.

## 5 EXPERIMENTS

### 5.1 EXPERIMENTAL SETUP

**Datasets:** We generated 10 synthetic datasets ($p = 10$) as was used in Tsang et al. (2018a) to test the accuracy of different interaction detection methods. We added a scaled Gaussian noise $\eta \sim 0.1 \times N(0, 1)$ to get the noisy data. We also selected 5 real datasets, namely the *Elevators* for controlling an F16 aircraft (Itorgo, 2019), *Parkinsons* for predicting the total UPDRS scores (Tsanas et al., 2009), *Skillcraft* for game player behavior analysis (Thompson et al., 2013), *Cal housing* for house price prediction (Pace & Barry, 1997), and *Bike sharing* for predicting hourly bike rental count (Fanaee-T & Gama, 2014). The datasets are preprocessed, and details are shown in supplement D.

**Neural Network Configuration:** For the baseline OverparaFC (Teacher), we used the architecture of $p$-5000-900-400-100-30-1 (indicating a deep neural network with $p$ inputs, one output, and 5 hidden layers with 5000, 900, 400, 100, 30 neurons, respectively), which is a standard ReLU network with the last hidden layer being linear. For the ParaACE model (Student), the size of the network mainly depends on the number of candidate interactions we choose; more interactions imply more blocks. However, the configuration for each subnetwork is fixed, the structure of which is 1-50-8-1 for main effects and 2-50-8-1 for pairwise interactions. These subnetworks are ReLU networks with linear output. The fix-up layer is chosen as a single layer ResNet with the number of neurons equal to 15. All the above networks were initialized with Kaiming's strategy (He et al., 2015).

**Optimizer Setup:** We chose adam (Kingma & Ba, 2014) as the optimizer, with the default setup in MXNet (Chen et al., 2015), and the batch size is set to be 500 for all the datasets.

### 5.2 PAIRWISE INTERACTION DETECTION

We first test on 10 different synthetic datasets ($p = 10$) (Tsang et al., 2018a) to show the superiority of our proposed method. Datasets with higher input dimensions ($p = 50, 100$) were compared as well, see supplement F. To compare the detection performance of different methods, we adopt the ROC-AUC metric (see supplement E). In Table 1, we compared our proposed method with various existing methods, including ANOVA (Wonnacott & Wonnacott, 1990), AG (Sorokina et al., 2008), IH (Janizek et al., 2020), NID (Tsang et al., 2018a), and PID (Liu et al., 2020). HierLasso (Bien et al., 2013) and RuleFit (Friedman et al., 2008) have been omitted due to their weak performances. For our method, we pick the same number of data samples $n = 10000$ for the training of OverparaFC and run 5 independent Monte Carlo tests. In Algorithm 1, we pull each arm 3 times for initialization. Each arm will be pulled at maximally $m = 100$ times, and we terminate when $k = 20$ strongest interactions stand out. We take $\hat{\mu}_r(t)$ upon termination as the final scores for our method. In this experiment, our proposed method needs around 1500 pulls of arms to pick out the top 20 interactions, however, naively pulling each arm 100 times needs $100C_{10}^2 = 4500$ pulls in total.

Table 1: Interaction detection methods in comparison. ROC-AUCs of pairwise interaction strengths proposed by us (with $h_{i(j)} = 0.8$) versus various baselines on a test suite of synthetic functions (see Table 4 in the supplement).

| | ANOVA[a,b] | AG | IH[a] | NID[a] | PID[a] | Our method[a,b] |
|---|---|---|---|---|---|---|
| $F_1(\mathbf{x})$ | 0.992 | $\mathbf{1 \pm 0.0}$ | 0.989 | $0.970 \pm 9.2e{-}3$ | $0.986 \pm 4.1e{-}3$ | $0.947 \pm 2.5e{-}2$ |
| $F_2(\mathbf{x})$ | 0.468 | $0.88 \pm 1.4e{-}2$ | **0.968** | $0.79 \pm 3.1e{-}2$ | $0.804 \pm 5.7e{-}2$ | $0.944 \pm 3.2e{-}2$ |
| $F_3(\mathbf{x})$ | 0.657 | $\mathbf{1 \pm 0.0}$ | 1 | $0.999 \pm 2.0e{-}3$ | $\mathbf{1 \pm 0.0}$ | $\mathbf{1 \pm 0.0}$ |
| $F_4(\mathbf{x})$ | 0.563 | $0.999 \pm 1.4e{-}3$ | 1 | $0.85 \pm 6.7e{-}2$ | $0.935 \pm 3.9e{-}2$ | $\mathbf{0.997 \pm 2.7e{-}3}$ |
| $F_5(\mathbf{x})$ | 0.544 | $0.67 \pm 5.7e{-}2$ | 1 | $\mathbf{1 \pm 0.0}$ | $\mathbf{1 \pm 0.0}$ | $0.988 \pm 2.3e{-}2$ |
| $F_6(\mathbf{x})$ | 0.780 | $0.64 \pm 1.4e{-}2$ | 0.932 | $0.98 \pm 6.7e{-}2$ | $\mathbf{1 \pm 0.0}$ | $0.925 \pm 3.7e{-}2$ |
| $F_7(\mathbf{x})$ | 0.726 | $0.81 \pm 4.9e{-}2$ | 0.611 | $0.84 \pm 1.7e{-}2$ | $\mathbf{0.888 \pm 2.8e{-}2}$ | $0.714 \pm 7.0e{-}2$ |
| $F_8(\mathbf{x})$ | 0.929 | $0.937 \pm 1.4e{-}3$ | 0.954 | $0.989 \pm 4.4e{-}3$ | $\mathbf{1 \pm 0.0}$ | $0.984 \pm 4.2e{-}3$ |
| $F_9(\mathbf{x})$ | 0.783 | $0.808 \pm 5.7e{-}3$ | 0.831 | $0.83 \pm 5.3e{-}2$ | $\mathbf{0.972 \pm 2.9e{-}2}$ | $0.948 \pm 4.6e{-}2$ |
| $F_{10}(\mathbf{x})$ | 0.765 | $\mathbf{1 \pm 0.0}$ | 1 | $0.995 \pm 9.5e{-}3$ | $0.987 \pm 3.5e{-}2$ | $\mathbf{1 \pm 0.0}$ |
| average | 0.721 | $0.87 \pm 1.4e{-}2$ | 0.931 | $0.92 \pm 2.3e{-}2$ | $\mathbf{0.957 \pm 6.2e{-}2}$ | $0.945 \pm \mathbf{5.6e{-}3}$[1] |
| Time (s) | - | - | 132 | 0.007 | - | 14.7 |

[1] Small standard deviation means high stability, [a] post-hoc method, [b] model agnostic method.

From the results, we can conclude the following: (1) PID outperforms all others; (2) our proposed method is as competitive as the PID method; (3) while the average performance of our method is only slightly worse than the PID method due to the poor result on dataset 7; this is due to the term $\frac{1}{1+(x_4x_5x_6x_7x_8)^2}$ in $F_7(\mathbf{x})$ has little impact on the output, and such interactions are hard to detect. Apart from the above facts, both NID and PID work by analyzing the connectivity of NN, which strongly relies on specific sparse ReLU networks and empirical interaction strength computations, and thus are risky to generalize and may fail for some other datasets. For instance, for the function $F_{11}(\mathbf{x}) = x_1^2 x_2 x_3^2 x_4 + x_5^2 x_6 x_7^2 x_8$, the ROC-AUC of NID degrades to around 0.69, while our method maintains 1.0 robustly. Moreover, our method is model-agnostic and can be directly applied to any pre-trained learning models, such as tree models, kernel regression models, etc., which is impossible for other methods.

## 5.3 MODEL COMPRESSION WITH INTERACTION KNOWLEDGE

In this section, we construct a Student model with the learned interactions. With the interaction knowledge extracted from the Teacher model, we first show that the Student model trained under the same setup can achieve well-improved prediction accuracy (on average). We use normalized RMSE (NRMSE), which is defined by $\frac{\text{RMSE}}{y_{\max} - y_{\min}}$ to measure the performances of all competing models such that the comparison across different datasets is fair and convenient.

For synthetic data, we use 800 training samples and 200 test samples. The noise $\eta$ is injected as described earlier. We choose the top 20 pairwise interactions with our new detection method to build the ParaACE model. We compared our proposed ParaACE with OverparaFC, KD for regression (Chen et al., 2017), and two recent pruning methods known as Lottery Ticket Hypothesis (LTH) (Frankle & Carbin, 2019) and SynFlow (Tanaka et al., 2020). The results are shown in Table 2. As we expected, the KD method shows similar performance as the OverparaFC, because the Student model tries hard to approximate its Teacher. However, our ParaACE model can improve the performance of the OverparaFC by 26.0% on average, and the model compression ratio (CR := $\frac{\text{\# parameters in the uncompressed model}}{\text{\# parameters in the compressed model}}$) is around 300. More details can be found in supplement L. We also found that the ParaACE model is sample efficient. To be concrete, the ParaACE trained on a small training set can surpass the over-parameterized NN trained with substantially more data, see Figure 13 in supplement K.

Table 2: NRMSE of the baselines: OverparaFC, KD, and LTH versus our proposed ParaACE tested on the synthetic datasets in Table 4. (The results were averaged over 5 folds.)

|  | $F_1$ | $F_2$ | $F_3$ | $F_4$ | $F_5$ | $F_6$ | $F_7$ | $F_8$ | $F_9$ | $F_{10}$ | Average | CR |
|---|---|---|---|---|---|---|---|---|---|---|---|---|
| OverparaFC | 0.026 | 0.054 | 0.059 | 0.062 | 0.042 | 0.041 | 0.018 | 0.024 | 0.032 | 0.023 | 0.038 | 1 |
| KD (student) | 0.029 | 0.055 | 0.061 | 0.064 | 0.041 | 0.042 | 0.019 | 0.024 | 0.032 | 0.024 | 0.039 | 278 |
| LTH | 0.027 | 0.044 | 0.032 | 0.032 | 0.039 | 0.033 | 0.018 | 0.021 | 0.029 | 0.025 | 0.030 | 18 |
| SynFlow | 0.025 | 0.048 | 0.034 | 0.035 | 0.039 | 0.031 | **0.015** | **0.018** | 0.026 | 0.022 | 0.029 | 280 |
| **ParaACE** | **0.025** | **0.035** | **0.031** | **0.030** | **0.038** | **0.025** | 0.016 | 0.019 | **0.023** | **0.021** | **0.026** | **283** |

For real-world datasets, we built the ParaACE model with the top 50 pairwise interactions for Elevators, Parkinsons, Skillcraft, Bike sharing, and the top 20 pairwise interactions for Cal housing (since the number of features is smaller). For most datasets, the detected interactions can help improve the predictive performance and significantly reduce the model size, see Table 3 for detailed results. The comparison with KD is omitted as it is supposed to be close to the Teacher.

Table 3: Performance comparison between OverparaFC, LTH, SynFlow, and ParaACE on real-world datasets. (The results were averaged over 5 folds.)

| Datasets | $N$ | $p$ | OverparaFC NRMSE | Parameters | LTH NRMSE | CR | SynFlow NRMSE | CR | **ParaACE** NRMSE | Parameters | CR |
|---|---|---|---|---|---|---|---|---|---|---|---|
| Elevators | 16599 | 18 | 0.0483 | 4999461 | 0.0523 | 87 | 0.0479 | 120 | **0.0475** | 39848 | **125** |
| Parkinsons | 5875 | 20 | 0.0251 | 5009461 | **0.0180** | 87 | 0.0229 | 120 | 0.0204 | 40946 | **122** |
| Skillcraft | 3338 | 19 | 0.0937 | 5004461 | 0.1228 | 87 | 0.0968 | 120 | **0.0929** | 40397 | **124** |
| Bike sharing | 17379 | 15 | **0.0403** | 4984461 | 0.0404 | 87 | 0.0405 | 120 | 0.0420 | 38201 | **130** |
| Cal housing | 20640 | 8 | 0.1059 | 4949461 | 0.1038 | 87 | 0.1026 | 120 | **0.1022** | 16388 | **302** |

Due to space limitations, we give more results for classification tasks in supplement I. Furthermore, we show the performance gain induced by interaction detection via ablation study in supplement N.

### 5.4 MODEL INTERPRETATION

For synthetic data, we take $F_{10}(\mathbf{x}) = \sinh(x_1 + x_2) + \arccos(\tanh(x_3 + x_5 + x_7)) + \cos(x_4 + x_5) + \sec(x_7 x_9)$ as an example. Three dominant pairwise interactions $\{x_1, x_2\}, \{x_4, x_5\}, \{x_7, x_9\}$ have been detected successfully and visualized in Figure 4 (left). The ground truth transformations are shown in the upper part of the figure. Besides, the transformations for the single features can also be learned accurately. Taking $F_3(\mathbf{x}) = \exp|x_1 - x_2| + |x_2 x_3| - (x_3^2)^{|x_4|} + \log(x_4^2 + x_5^2 + x_7^2 + x_8^2) + x_9 + \frac{1}{1+x_{10}^2}$ for example, wherein feature $x_6$ is not involved, $x_9$ is linearly transformed, and $x_{10}$ is transformed by an even function. All of them are correctly learned as shown in Figure 4 (right).

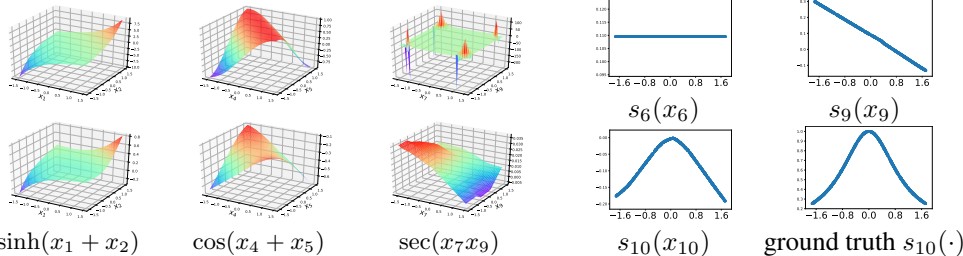

$$\sinh(x_1 + x_2) \qquad \cos(x_4 + x_5) \qquad \sec(x_7 x_9) \qquad s_{10}(x_{10}) \qquad \text{ground truth } s_{10}(\cdot)$$

Figure 4: Visualization of the transformations in $F_{10}$ (left) and $F_3$ (right).

For real applications, we first take the Cal housing dataset for further analysis as less expert knowledge is required. From Figure 5, we can easily recognize that {*longitude, latitude*} and {*totalRooms,totalBedrooms*} show strong interactions. This makes sense, because {*longitude, latitude*} indicates the location, and {*totalRooms,totalBedrooms*} indicates the fraction of bedrooms. Also in the Parkinsons dataset, we find {*Age, Sex*} are interacted, which is consistent with the result that "age at onset (of Parkinson's disease) was 2.1 years later in women than in men" in Haaxma et al. (2007). Interaction detection may also have a significant impact on drug combination discovery (Julkunen et al., 2020). We applied our interaction detection method on an RF model trained with the Drug combination dataset. Among the top 34 detected drug-drug interactions, 15 pairwise interactions have been verified in the DrugBank database (Wishart et al., 2018) (see supplement J). These results confirmed the effectiveness of our interaction detection method and strongly call for careful investigation of the unexplored drug pairs, potentially of great value.

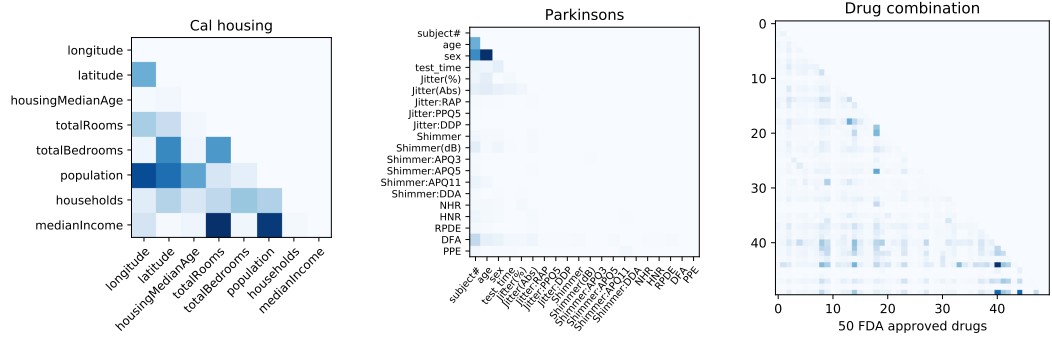

Figure 5: Heat maps of interaction strengths for Cal housing, Parkinsons, and Drug combination data. Note that darker color indicates stronger interaction.

## 6 CONCLUSION

This paper proposed a fast, generic, and model-agnostic interaction detection method. A thorough analysis of the computational complexity of our adaptive method proved its superior efficacy. The detected interaction knowledge can be further exploited to build a novel Student model, called ParaACE, which is compact, more interpretable, and mostly more accurate than its Teacher and other competing methods. We also showed experimentally that our ParaACE model is sample efficient owing to its simple architecture. Its model-agnostic property enables tremendous applications in different domains of finance, medicine, biology, wireless communication, where a significant number of well-performed black-box models have been built but not interpreted yet.

ACKNOWLEDGMENTS

This work was supported by Guangdong Provincial Key Laboratory of Big Data Computing and by NSFC under Grant 61731018. The corresponding author is Feng Yin.

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

# Supplementary Document
# Fast Generic Interaction Detection for Model Interpretability and Compression

## A    PROBLEMS WITH ANALYTICAL EVALUATION

In this section, we first illustrate the conceptual difference between local and global interaction and then state the problems of analytical evaluation.

**Illustration of Local vs. Global Interaction.**  A quick example showing the difference of local/global interaction is the MATLAB symbol[1] (Figure 6(a)). There is no local interaction inside the flat region, but two input variables globally interact.

Analytical evaluation works for neural networks with differentiable activation functions, *e.g.*, `sigmoid`, `tanh`, etc. However, it is problematic when we use neural networks with piece-wise linear activation functions (PLNN), such as `ReLU`, `Leaky ReLU`, and so on. The Hessian will be a zero matrix at every second-order differentiable point, and it will no longer provide information about local interaction. Imagine the function landscape is joint with many facets, and the interaction information is hidden in the boundary of those facets (the set of non-differentiable points).

A simple example is checkerboard-like function, *e.g.*, XOR function $F(x_1, x_2) = \mathbb{1}\{x_1 x_2 > 0\}$, where $x_1, x_2$ are uniformly drawn from $[-1, 1]$. In this case, $x_1$ and $x_2$ are clearly interacted, however, the second-order derivatives $\frac{\partial F}{\partial x_1 \partial x_2}$ will be zero for almost all sampled points. The interaction information is hidden at the line $x_1 = 0$ and $x_2 = 0$.

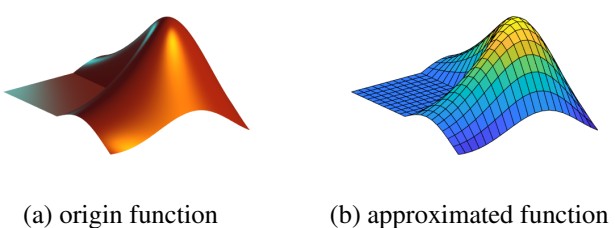

(a) origin function                    (b) approximated function

Figure 6: The MATLAB symbol. (a)Imagine the MATLAB symbol as the landscape of a function $F(x_1, x_2)$, there is no interaction locally in the flat region, but $x_1$ and $x_2$ interact globally, i.e., there is no global decomposition as $F(x_1, x_2) = a(x_1) + b(x_2)$. (b) (Problems with analytical evaluation) The function is approximated by a PLNN, and the landscape is spliced by flat facets. The Hessian matrix is a zero matrix on almost every point (no local interaction), but $x_1$ and $x_2$ still interact globally.

## B    PROOF OF THEOREM 3.1 AND DISCUSSION ON THE CHOICE OF $h$

**Theorem 3.1.**   *For any $x$ and $y$, function $F$ shows no interaction between them, i.e., it can be decomposed as $F(x, y) = a(x) + b(y)$ **if and only if**, $\forall\, h, k > 0$, $F(x + h, y + k) - F(x + h, y - k) - F(x - h, y + k) + F(x - h, y - k) = 0$.*

*Proof.*  ($\Rightarrow$) is trivial. ($\Leftarrow$) We will directly have, $\forall h, k > 0$ and $\forall x, y$,

$$\frac{F(x + h, y + k) - F(x + h, y - k)}{k} = \frac{F(x - h, y + k) - F(x - h, y - k)}{k},$$

$$\frac{F(x + h, y + k) - F(x - h, y + k)}{h} = \frac{F(x + h, y - k) - F(x - h, y - k)}{h}.$$

---

[1]With permission of MathWorks.

Let $k \to 0$, we will have $\frac{\partial F}{\partial y}(x+h, y) = \frac{\partial F}{\partial y}(x-h, y)$ for any positive $h$; let $h \to 0$, we will have $\frac{\partial F}{\partial x}(x, y+k) = \frac{\partial F}{\partial x}(x, y-k)$ for any positive $k$. Thus, $\frac{\partial F}{\partial y}$ is irrespective of $x$ and $\frac{\partial F}{\partial x}$ is irrespective of $y$. □

**Remark B.1.** *Theorem 3.1 can be extended for higher-order interaction with the same machinery.*

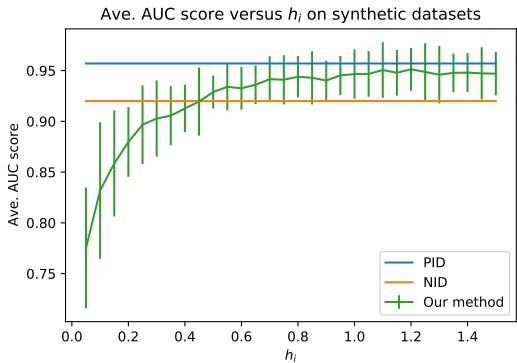

Figure 7: Average ROC-AUC score versus $h_{i(j)}$ on synthetic datasets. The error bar represents one standard deviation (5 folds).

**A guide for the choice of $h_{i(j)}$.** We empirically test the sensitivity of $h_{i(j)}$, see Figure 7. For *standardized features*, we suggest using $h_{i(j)} \in [0.6, 0.9]$ as the rule-of-thumb.

**Further discussions on the choice of $h_{i(j)}$.** People may argue that for the interaction detection task, a small number of evaluations for each feature pair are already enough to get pretty good results, and the benefits of using the UCB algorithm is trivial. This is not always true. We address this from two perspectives in the following. First, in many scenarios, the function can only be evaluated sequentially, and the evaluation may be very expensive. The benefits of the UCB algorithm are revealed then. Second, we empirically study the performance of evaluating each arm evenly under different configurations (perturbation size $h_{i(j)}$, and the number of evaluations for each arm), see Figure 8. The ROC-AUC scores were obtained from the OverparaFC model trained as described in Section 5.2. We expect that given the fixed perturbation size $h$, the ROC-AUC scores will rise as the number of evaluations increases. This is consistent with the observation that the bricks are brighter from top to bottom. Also, we observed that with a properly chosen $h \in [0.6, 0.9]$, the interactions can be correctly detected with a relatively small number of evaluations (the bricks are brighter in the middle, see dataset 3, 4, and 5). The corresponding reasons can be that (a) if $h$ is too small, four evaluated points lie in the same flat region with high probability, which is harmful to interaction detection; (b) if $h$ is too large, some evaluated points may be out of the distribution of the training data, which makes the detected interaction strength less representative. Even though the suggestion of using $h_{i(j)} \in [0.6, 0.9]$ is purely empirical, the plots here help us better understand the essence of statistical interactions and their detection process.

## C PROOF OF THEOREM 3.2

Recall that we construct the $1 - \delta$ confidence intervals of $\hat{f}_\ell(r)$ as,

$$C(\ell) = \begin{cases} \sqrt{\frac{2\sigma^2 \log \frac{2}{\delta}}{\ell}} & \text{if } \ell \leq m \\ 0 & \text{if } \ell > m \end{cases},$$

$$\mu_r \in \left[ \hat{f}_\ell(r) - C(\ell), \hat{f}_\ell(r) + C(\ell) \right], \text{ w.p. } 1 - \delta.$$

This is derived from concentration inequality (Inequality 2.9 in Wainwright (2019)). We set $\delta = \frac{2}{n^3 m}$, which is a small number and easy to analyze. We have,

**Lemma C.1** (high probability event/clean event). *With probability (w.p.) $1 - \frac{2}{n^2}$, all true values $\mu_r$ lie in the their confidence intervals during the run of the algorithm.*

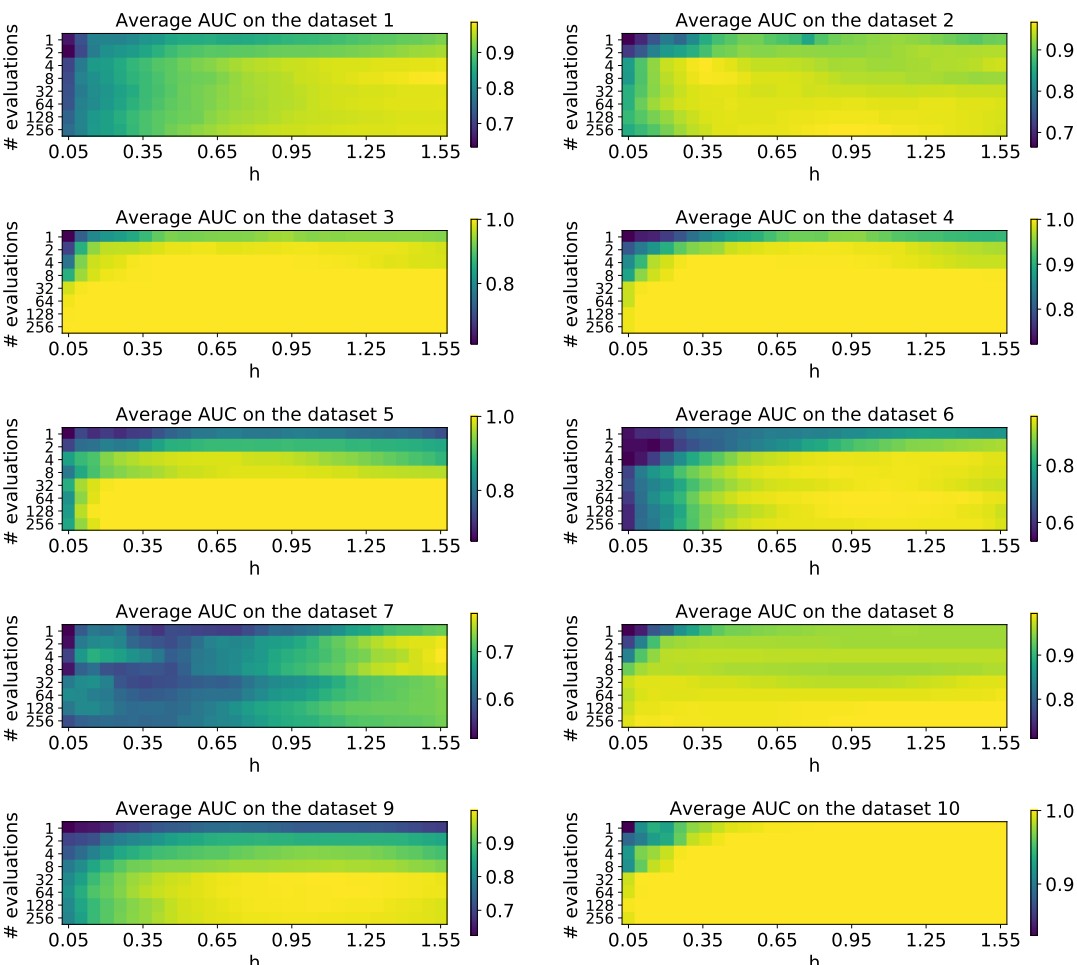

Figure 8: The ROC-AUC scores obtained for 10 synthetic datasets under different configurations of perturbation size $h$ (horizontal axis), and the number of evaluations on each interaction pair (vertical axis). We expect that given the fixed perturbation size $h$, the ROC-AUC scores will rise as the number of evaluations increases. This is consistent with the observation that the bricks are brighter from top to bottom. Also, we observed that with a properly chosen $h \in [0.6, 0.9]$, the interactions can be correctly detected with a relatively small number of evaluations (the bricks are brighter in the middle, see dataset 3, 4, and 5).

*Proof.* It's equivalent to proof:

$$P(\{ \forall\, t, \forall\, r \in [n], |\mu_r - \hat{\mu}_r(t)| \le C_r(t)\}) \ge 1 - \frac{2}{n^2}$$

The opposite of this event is

$$\exists\, t \text{ and } r,\ \text{s.t.,}\ |\mu_r - \hat{\mu}_r(t)| > C_r(t).$$

Since we have,

$$P(\{|\mu_r - \hat{\mu}_r(t)| > C_r(t)\}) \le \delta = \frac{2}{n^3 m},$$

and we evaluate at most $nm$ times ($n$ arms, each arm is pulled $m$ times),

$$P(\{\exists\, t \text{ and } r,\ \text{s.t.,}\ |\mu_r - \hat{\mu}_r(t)| > C_r(t)\}) \le \delta nm = \frac{2}{n^2} \quad \textit{(union bound)}$$

$$P(\{ \forall\, t, \forall\, r \in [n], |\mu_r - \hat{\mu}_r(t)| \le C_r(t)\}) \ge 1 - \frac{2}{n^2} \quad \textit{(clean event)}$$

$\square$

Using Lemma C.1, we restate Theorem 3.2 as follows.

**Theorem C.2** (Restating Theorem 3.2). *With probability $1 - \Theta(\frac{1}{n^2})$, Algorithm 1 returns the $k$-strongest interactions with at most*

$$M \leq \sum_{i=1}^{n} \left( \left( \frac{24\sigma^2}{\Delta_i^2} \log(nm) \right) \wedge m \right)$$

*local interaction strength evaluations. This takes $\mathcal{O}\left( \sum_{i=1}^{n} \log(n) \left( \left( \frac{\sigma^2 \log(nm)}{\Delta_i^2} \right) \wedge m \right) \right)$ time.*

*Proof.* We first analyze the running of algorithm till the first strongest interaction comes out. If we choose to update arm $i \neq i_1^*$ at time $t$, then we have

$$\hat{\mu}_i(t) + C_i(t) \geq \hat{\mu}_{i_1^*}(t) + C_{i_1^*}(t). \tag{6}$$

For equation 6 to occur, at least one of the following three events must occur:

$$\mathcal{E}_1 = \left\{ \hat{\mu}_{i_1^*}(t) \leq \mu_{i_1^*} - C_{i_1^*}(t) \right\},$$
$$\mathcal{E}_2 = \left\{ \hat{\mu}_i(t) \geq \mu_i + C_i(t) \right\},$$
$$\mathcal{E}_3 = \left\{ \Delta_i^{(1)} = \mu_{i_1^*} - \mu_i \leq 2C_i(t) \right\}.$$

To see this, note that if none of $\mathcal{E}_1, \mathcal{E}_2, \mathcal{E}_3$ occurs, we have

$$\hat{\mu}_i(t) + C_i(t) \overset{(\neg \mathcal{E}_2)}{<} \mu_i + 2C_i(t) \overset{(\neg \mathcal{E}_3)}{<} \mu_{i_1^*} \overset{(\neg \mathcal{E}_1)}{<} \hat{\mu}_{i_1^*} + C_{i_1^*}(t).$$

From Lemma C.1, $\mathcal{E}_1$ and $\mathcal{E}_2$ do not occur during the run of the algorithm with probability $1 - \frac{2}{n^2}$, because

$$\text{w.p. } \left( 1 - \frac{2}{n^2} \right) : |\mu_i - \hat{\mu}_i(t)| \leq C_i(t), \forall\, i \in [n],\, \forall\, t. \tag{7}$$

It also implies w.p. $1 - \Theta(\frac{1}{n^2})$, the algorithm does not stop pulling arm $i$ until event $\mathcal{E}_3$ stops occurring.

Let $\zeta_i^{(w)}$ index the iteration in which Algorithm 1 evaluates arm $\mathcal{A}_i$ for the last time before declaring it to be the $w$-th strongest interaction. Let's first consider how many pulls are needed for each arm to pick out the first strongest interaction. If $C_i(\zeta_i^{(1)}) \leq \frac{\Delta_i^{(1)}}{2}$ for arm $\mathcal{A}_i$, then we can stop evaluating arm $\mathcal{A}_i$, that is,

$$\frac{\Delta_i^{(1)}}{2} \geq \sqrt{\frac{2\sigma^2 \log n^3 m}{T_i(\zeta_i^{(1)})}} \text{ or } C_i(\zeta_i^{(1)}) = 0,$$

$$\implies T_i(\zeta_i^{(1)}) \geq \frac{8\sigma^2}{(\Delta_i^{(1)})^2} \log(n^3 m) \text{ or } T_i(\zeta_i^{(1)}) \geq m.$$

Note that $T_i(t)$ denotes for the number of pulls of arm $\mathcal{A}_i$ at iteration $t$. We will evaluate arm $\mathcal{A}_i$ at most $\frac{8\sigma^2}{(\Delta_i^{(1)})^2} \log(n^3 m) \wedge m$ times. To pick out the first strongest interaction, we need at most $M$ evaluations satisfying

$$M \leq \sum_{i=1}^{n} \left( \left( \frac{8\sigma^2}{(\Delta_i^{(1)})^2} \log(n^3 m) \right) \wedge m \right).$$

This result can be extend to find all top $k$ strongest interactions concretely,

$$M \leq \sum_{i=1}^{n} \left( \left( \frac{8\sigma^2}{(\Delta_i^{(k)})^2} \log(n^3 m) \right) \wedge m \right),$$

$$= \sum_{i=1}^{n} \left( \left( \frac{24\sigma^2}{(\Delta_i^{(k)})^2} \log(n \sqrt[3]{m}) \right) \wedge m \right),$$

$$\leq \sum_{i=1}^{n} \left( \left( \frac{24\sigma^2}{(\Delta_i^{(k)})^2} \log(nm) \right) \wedge m \right).$$

The Algorithm 1 takes $\mathcal{O}\left( \sum_{i=1}^{n} \log(n) \left( \left( \frac{\sigma^2 \log(nm)}{\Delta_i^2} \right) \wedge m \right) \right)$ time, where $\mathcal{O}(\log(n))$ is the time for maintaining a priority queue(to find the minimal LCB or maximal UCB) in each iteration. $\square$

**Theorem C.3** (Remark 3.1). *If we further assume that $\Delta_i \sim \mathcal{N}(\gamma, 1)$, for some $\gamma$, and $m = cn$, for $c \in [0, 1]$, then the expected pulls of arms (over randomness in $\Delta_i$) satisfies*

$$\mathbb{E}[M] \leq \mathcal{O}(n \log(nm) + km)$$

*with probability $1 - \Theta(\frac{1}{n^2})$ over randomness in Algorithm 1. Thus, the expected running time is well bounded by $\mathcal{O}(n \log(nm) \log(n) + km \log(n))$.*

*Proof.* This follows directly from the Appendix 2 of Bagaria et al. (2018a). $\square$

## D  DATASETS PRE-PROCESSING

### D.1  SYNTHETIC TEST SUITE

To make our experimental results convincing, we follow the test suite ever used in Tsang et al. (2018a) with the details given in Table 4. Among others, $F_1$ is a widely used test function for interaction detection, which can be generated as described in Sorokina et al. (2008). For all the other functions, the input dimension is set to 10, and $x_1, \ldots, x_{10} \overset{i.i.d.}{\sim} U(-1, 1)$.

Table 4: Test suite of data-generating functions

| | |
|---|---|
| $F_1(\mathbf{x})$ | $\pi^{x_1 x_2} \sqrt{2x_3} - \sin^{-1}(x_4) + \log(x_3 + x_5) - \frac{x_9}{x_{10}} \sqrt{\frac{x_7}{x_8}} - x_2 x_7$ |
| $F_2(\mathbf{x})$ | $\pi^{x_1 x_2} \sqrt{2|x_3|} - \sin^{-1}(0.5 x_4) + \log(|x_3 + x_5| + 1) + \frac{x_9}{1 + |x_{10}|} \sqrt{\frac{x_7}{1 + |x_8|}} - x_2 x_7$ |
| $F_3(\mathbf{x})$ | $\exp|x_1 - x_2| + |x_2 x_3| - (x_3^2)^{|x_4|} + \log(x_4^2 + x_5^2 + x_7^2 + x_8^2) + x_9 + \frac{1}{1 + x_{10}^2}$ |
| $F_4(\mathbf{x})$ | $\exp|x_1 - x_2| + |x_2 x_3| - (x_3^2)^{|x_4|} + (x_1 x_4)^2 + \log(x_4^2 + x_5^2 + x_7^2 + x_8^2) + x_9 + \frac{1}{1 + x_{10}^2}$ |
| $F_5(\mathbf{x})$ | $\frac{1}{1 + x_1^2 + x_2^2 + x_3^2} + \sqrt{\exp(x_4 + x_5)} + |x_6 + x_7| + x_8 x_9 x_{10}$ |
| $F_6(\mathbf{x})$ | $\exp(|x_1 x_2| + 1) - \exp(|x_3 + x_4| + 1) + \cos(x_5 + x_6 - x_8) + \sqrt{x_8^2 + x_9^2 + x_{10}^2}$ |
| $F_7(\mathbf{x})$ | $(\arctan(x_1) + \arctan(x_2))^2 + \max(x_3 x_4 + x_6, 0) - \frac{1}{1 + (x_4 x_5 x_6 x_7 x_8)^2} + \left( \frac{|x_7|}{1 + |x_9|} \right)^5 + \sum_{i=1}^{10} x_i$ |
| $F_8(\mathbf{x})$ | $x_1 x_2 + 2^{x_3 + x_5 + x_6} + 2^{x_3 + x_4 + x_5 + x_7} + \sin(x_7 \sin(x_8 + x_9)) + \arccos(0.9 x_{10})$ |
| $F_9(\mathbf{x})$ | $\tanh(x_1 x_2 + x_3 x_4) \sqrt{|x_5|} + \exp(x_5 + x_6) + \log((x_6 x_7 x_8)^2 + 1) + x_9 x_{10} + \frac{1}{1 + |x_{10}|}$ |
| $F_{10}(\mathbf{x})$ | $\sinh(x_1 + x_2) + \arccos(\tanh(x_3 + x_5 + x_7)) + \cos(x_4 + x_5) + \sec(x_7 x_9)$ |

### D.2  REAL DATASETS

All the real datasets are publicly available. We preprocessed the datasets as follows. For Parkinsons data, we remove the column *motor UPDRS* and predict the *total UPDRS*. For SkillCraft data, the

target is set to be *LeagueIndex*. For the Bike sharing data, the feature *weather* is converted into a one-hot representation. Before feeding the data into neural networks, we performed data normalization first for all datasets.

The Drug combination dataset is processed as follows. There are 110 features extracted, among which the first 50 features are the concentration of 50 unique FDA-approved drugs, and the last 60 features are the one-hot encodings for the cell lines.

## E   ROC-AUC FOR PAIRWISE INTERACTION DETECTION

We calculate the ROC-AUC scores for the synthetic datasets, where the ground truth interaction pairs are known. To obtain the ROC-AUC value, two vectors are needed, namely the *pairwise interaction score* $\in \mathbb{R}_+^{45}$ and *ground truth* $\in \{0, 1\}^{45}$, both of them are of dimension $\frac{p(p-1)}{2} = 45$. By setting different thresholds for the pairwise interaction score ranking, different classifiers with False Positive (FP) rate and True Positive (TP) rate can be obtained. The ROC-AUC value is approximated from those (FP, TP) pairs by the trapezoidal rule.

## F   PAIRWISE INTERACTION DETECTION ON HIGH- DIMENSIONAL DATASETS

We created two new high-dimensional datasets by simply combining the 10 synthetic datasets considered in the paper. The new datasets have either 50 (using F1-F5) or 100 features (using F1-F10). The labels of the two datasets are the sum of the original labels in each dataset of input dimension $p = 10$. The training sample size is increased to 500,000 to mimic big data. We compared 5 different model architectures and our method consistently outperforms NID, see the Table 5 below for the results.

Table 5: ROC-AUCs comparison for high-dimensional datasets

| ROC-AUC score (NID/Ours) | data size: 500,000 * 100 | data size: 500,000 * 50 |
|---|---|---|
| Big network[1] with main effect + L1reg | 0.743/**0.768** | 0.831/**0.864** |
| Big network without main effect + L1reg | 0.699/**0.700** | 0.803/**0.855** |
| Small network[2] with main effect + L1reg | 0.731/**0.744** | 0.836/**0.859** |
| Small network without main effect + L1reg | 0.701/**0.742** | 0.796/**0.840** |
| Standard big network without main effect | 0.646/**0.653** | 0.583/**0.793** |

[1] Big network: p-5000-900-400-100-30-1
[2] Small network: p-140-100-60-20-1

Here we also report the computational complexity of our UCB-based interaction detection method. Due to the randomness of the UCB algorithm, the number of pulls fluctuates. Note that we select top $k = 200$ interaction pairs for 100-dimensional case, and the total number of pulls is around 40000, while the naive approach needs $100 \times 100 \times 99/2 = 495000$ pulls. We select top $k = 100$ interaction pairs for 50-dimensional case, and the total number of pulls is around 21000, while the naive approach needs $100 \times 50 \times 49/2 = 122500$ pulls.

**Discussion on High Dimensional Data:** For high dimensional data, the Hessian matrix will be too huge to handle. One possible solution is to focus on the important features. We may first take advantage of DeepLIFT(Shrikumar et al., 2017) or SHAP (Lundberg & Lee, 2017) to screen out the most important features, and then apply our interaction detection method. If the number of important features is still too large to handle, we may use ANOVA and F-test to select an affordable number of interaction candidates to pull $\log(n)$ times, and explore the remaining pairs with a small probability, say 5%, like the common strategy used in reinforcement learning..

## G   EXTENSION TO HIGHER-ORDER INTERACTIONS

We define three-way interaction as follows, and higher-order interactions can be defined similarly.

**Definition G.1** (three-way interaction). *A function $F : \mathbb{R}^p \to \mathbb{R}$ is said to exhibit an interaction among three of its variables $x_i$, $x_j$ and $x_k$ if,*

$$E_{\mathbf{x}} \left[ \frac{\partial^3 F(\mathbf{x})}{\partial x_i \partial x_j \partial x_k} \right]^2 > 0.$$

The naive method to detect higher-order interactions is quite similar to FANOVA graph (Muehlenstaedt et al., 2012). We give three examples of detecting higher-order interactions from the synthetic data. We detect the pairwise interaction strength first, then build a weighted graph (the edge weight corresponds to the interaction strength). By setting a proper number of clusters, higher-order interactions can be discovered correctly, see Figure 9.

(a) $y = x_0 x_1 + x_2 x_3 x_4 + x_5 x_6 x_7 + x_8 x_9$

(b) $y = e^{|x_0 - x_1|} * x_9 + |x_2 * x_3| - x_4^{2|x_5|} + (x_6 x_7)^2 + x_8$

(c) $y = x_0 x_1 + |x_1 + x_2 x_3| + x_4 x_5 - (x_5^2)^{x_6} - e^{x_7 + x_8 x_9}$

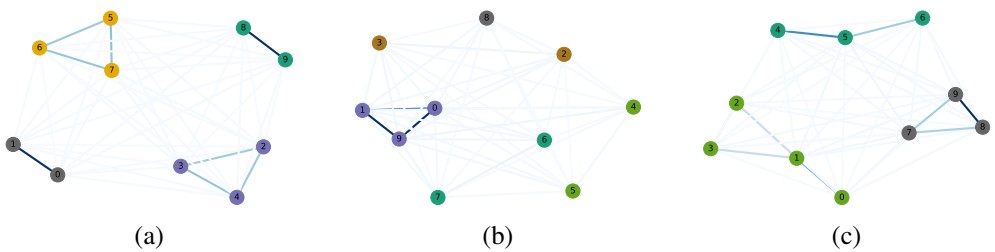

| (a) | (b) | (c) |

Figure 9: The nodes (features) are clustered correctly. The color of edges indicates pairwise interaction strength. Nodes with the same color belong to the same cluster.

---

**Algorithm 2** `Hierarchical Higher-Order Interaction Detection`

---

**Require:** The target number of $i$-way interactions $\{k^{(i)}\}_{i=2}^p$.
1: Detect top-$k^{(2)}$ pairwise interaction via UCB algorithm.
2: Construct an undirected graph $\mathcal{G}$ based on the detected pairwise interactions.
3: Enumerate all the cliques $\mathfrak{C}$ in the graph $\mathcal{G}$.
4: **for** $i = 3, 4, \cdots, p$ **do**
5:     **if** $\mathfrak{C}^{(i)} = \varnothing$ **then**
6:         **break**
7:     **end if**
8:     Find $\mathfrak{C}_{\text{refine}}^{(i)} = \{C \in \mathfrak{C}^{(i)} \mid$ every $i - 1$ complete subgraph of $C$ admits a detected interaction$\}$
9:     Detect the top $k^{(i)}$-strongest $i$-way interactions in $\mathfrak{C}_{\text{refine}}^{(i)}$ via the UCB algorithm.
10: **end for**
11: **return** All the detected interactions with their strengths.

---

One drawback of the above naive clustering method is that it doesn't allow for the overlap of nodes, i.e. one variable may appear in several interactions ($F_8$ in the test suite). To address the above problem, we propose the Algorithm 2, which detects the higher-order interactions hierarchically. We first construct an undirected graph $\mathcal{G}$ from the detected pairwise interactions. To shrink our search space for higher-order interactions, we restrict ourselves to the set of all cliques $\mathfrak{C}$ of the graph $\mathcal{G}$, where a clique $C \in \mathfrak{C}$ in a graph is a subset of vertices that are all joined by edges. We use $\mathfrak{C}^{(i)} \subseteq \mathfrak{C}$ to denote the collection of the $i$-cliques (the cliques with $i$ vertices). For example, to detect 3-way interactions, one only needs to search in the $\mathfrak{C}^{(3)}$, which contains the cliques with three vertices. Furthermore, we can shrink the search space to $\mathfrak{C}_{\text{refine}}^{(i)}$ for $i$-way interactions, based on the detected $(i - 1)$-way interactions (see line 8 in Algorithm 2). For example, while detecting 4-way interactions, we put 4-clique $\{2, 3, 4, 6\}$ into consideration only if all the 3-way interactions $\{3, 4, 6\}, \{2, 4, 6\}, \{2, 3, 6\}, \{3, 4, 6\}$ exist. We then use the UCB algorithm to verify if the cliques

in $\mathfrak{C}^{(i)}_{\text{refine}}$ admit the interaction relationship. In this way, we detect the $i$-way interactions from the detected $(i-1)$-way interactions information.

The 3-way interaction strength can be calculated from Equation 8. However, with the order of interaction increasing, the number of function evaluations for one gradient computation increases geometrically. We do not recommend detecting the interaction whose order is higher than four, which is uneconomical to compute and hard to interpret. We found our proposed Algorithm 2 can successfully detect the higher-order interactions for $F_8$ in the test suite [2].

$$
\begin{aligned}
\frac{\partial^2 F(\mathbf{x})}{\partial x_i \partial x_j \partial x_k} \approx \frac{1}{8h^3} \big[ &+ F(\mathbf{x} + h(\mathbf{e}_i + \mathbf{e}_j + \mathbf{e}_k)) + F(\mathbf{x} + h(-\mathbf{e}_i - \mathbf{e}_j + \mathbf{e}_k)) \\
&+ F(\mathbf{x} + h(\mathbf{e}_i - \mathbf{e}_j - \mathbf{e}_k)) + F(\mathbf{x} + h(-\mathbf{e}_i + \mathbf{e}_j - \mathbf{e}_k)) \\
&- F(\mathbf{x} + h(-\mathbf{e}_i + \mathbf{e}_j + \mathbf{e}_k)) - F(\mathbf{x} + h(\mathbf{e}_i - \mathbf{e}_j + \mathbf{e}_k)) \\
&- F(\mathbf{x} + h(\mathbf{e}_i + \mathbf{e}_j - \mathbf{e}_k)) - F(\mathbf{x} + h(-\mathbf{e}_i - \mathbf{e}_j - \mathbf{e}_k)) \big]
\end{aligned}
\tag{8}
$$

## H  MORE ON THE PARAMETRIC ACE

In section 4, we introduced a parametric ACE model based on the following generalized linear additive model with main effects and pairwise feature interactions, namely,

$$
\theta(Y) = \sum_{i=1}^{p} \beta_i s_i \left( x_{u_i}; \boldsymbol{\theta}_{s_i} \right) + \sum_{i=1}^{R} \beta_{p+i} r_i \left( \mathbf{x}_{\mathcal{I}_i}; \boldsymbol{\theta}_{r_i} \right) + \epsilon.
\tag{9}
$$

The consideration of interactions improves the model performance significantly, compare to Generalized Additive Neural Networks (GANN)(Potts, 1999; Agarwal et al., 2020)[3], see Table 6.

Table 6: NRMSE of the GANN versus our proposed ParaACE network on the synthetic datasets in (Table 4). (The results were averaged over 5 folds.)

|         | $F_1$ | $F_2$ | $F_3$ | $F_4$ | $F_5$ | $F_6$ | $F_7$ | $F_8$ | $F_9$ | $F_{10}$ | average | CR |
|---------|-------|-------|-------|-------|-------|-------|-------|-------|-------|----------|---------|-----|
| GANN    | 0.033 | 0.063 | 0.064 | 0.068 | 0.053 | 0.052 | 0.028 | 0.032 | 0.040 | 0.033    | **0.046** | **959** |
| ParaACE | 0.025 | 0.035 | 0.031 | 0.030 | 0.038 | 0.025 | 0.016 | 0.019 | 0.023 | 0.021    | **0.026** | **283** |

The idea behind the original nonparametric ACE algorithm (Breiman & Friedman, 1985) is to alternate between an inner process for finding the optimal transformation functions of the inputs and an outer process for finding the optimal transformation function of the output.

Here, we made several modifications. First, all the transformation functions are represented by small subnetworks. Second, we reformulate the above regression equation approximately as

$$
Y \approx \theta^{-1} \left( \sum_{i=1}^{p} \beta_i s_i \left( x_{u_i}; \boldsymbol{\theta}_{s_i} \right) + \sum_{i=1}^{R} \beta_{p+i} r_i \left( \mathbf{x}_{\mathcal{I}_i}; \boldsymbol{\theta}_{r_i} \right) \right).
\tag{10}
$$

Therefore, the fix-up layer shown in Figure 3 can be understood as the inverse optimal transformation of the output $\theta^{-1}(\cdot)$. To be more general, $\theta^{-1}(\cdot)$ takes all input transformations as individuals instead of their high-level summary (weighted sum). Our parametric ACE then alternately tunes the fix-up layer conditioned on the current optimal transformation layer (outer iteration) and tunes the optimal transformation layer conditioned on the current fix-up layer until some convergence condition is met.

Another important function of the fix-up layer is to alleviate the negative impacts of wrongly detected pairwise interactions and/or undetected higher-order interactions altogether on the output. For this purpose, we could make this fix-up layer a network of small subnetworks, so that the whole architecture is a network of small subnetworks. Each subnetwork can be regarded as a meta-neuron that is expected to be more competent than any SOTA activation function. With such a nice structure,

---

[2] A demo can be found at https://github.com/zhangtj1996/ParaACE.

[3] GANN is a parametric version of GAM, which only considers the transformations of univariates (single features).

we have divided the hyper-parameters naturally into blocks and the whole network can hopefully be made adaptive to new tasks more rapidly by tuning just a few blocks.

It is possible for ParaACE to learn "contradictory" models, which present the same function (Lengerich et al., 2020). For example, the main effect might be absorbed into the interaction effects. Thus the direct interpretation through subnetworks may not sound that convincing. Lengerich et al. (2020) proposed an algorithm to purify the interaction effect by calculating the functional ANOVA (Hooker, 2007), based on a piecewise-constant function $\hat{F}$. This algorithm is applicable to any function $F$ by first constructing a piecewise-constant approximation $\hat{F}$. Particularly, for our ParaACE model, the overall computation will be much easier since we explicitly have the univariate and bi-variate feature transformation. The number of bins required to approximate ParaACE with $\hat{F}$ reduces significantly, and so does the complexity of the follow-up purifying algorithm.

## I  EXTENSION TO CLASSIFICATION TASK

We generated the datasets for binary classification from the synthetic regression datasets (1000 samples with injected noise). We choose the median of the target as a threshold to separate each synthetic regression the dataset into two classes. The comparison between the baseline OverparaFC and our proposed ParaACE in terms of classification accuracy is shown in Table 7. For real-world datasets, we picked **Higgs Boson** (Baldi et al., 2014), **Spambase** (Dua & Graff, 2017), and **Diabetes** (Turney, 1994).Both the classification accuracy and the compression ratio are reported in Table 8.

Table 7: Accuracy of the baseline OverparaFC versus our proposed ParaACE network on the synthetic classification datasets. The results were averaged over 5 folds.

|  | $F_1$ | $F_2$ | $F_3$ | $F_4$ | $F_5$ | $F_6$ | $F_7$ | $F_8$ | $F_9$ | $F_{10}$ | Average | CR |
|---|---|---|---|---|---|---|---|---|---|---|---|---|
| OverparaFC | 0.955 | 0.896 | 0.887 | 0.872 | 0.907 | 0.939 | 0.964 | 0.971 | 0.932 | 0.953 | **0.927** | **1** |
| ParaACE | 0.949 | 0.908 | 0.94 | 0.946 | 0.916 | 0.94 | 0.96 | 0.935 | 0.919 | 0.942 | **0.936** | **283** |

Table 8: Performance comparison between OverparaFC and ParaACE on various real-world classification datasets.

| Datasets | $N$ | $p$ | OverparaFC Accuracy | Parameters | ParaACE Accuracy | CR |
|---|---|---|---|---|---|---|
| Higgs Boson | 98050 | 28 | $0.698 \pm 3.0e-3$ | 5049461 | **$0.730 \pm 2.8e-3$** | **184** |
| Spambase | 4601 | 57 | $0.950 \pm 8.0e-3$ | 5194461 | **$0.950 \pm 7.9e-2$** | **120** |
| Diabetes | 768 | 8 | $0.741 \pm 2.5e-2$ | 4949461 | **$0.789 \pm 1.7e-2$** | **302** |

## J  DETECTED PAIRWISE INTERACTIONS FOR SYNTHETIC AND REAL DATA

**Synthetic data:** Figure 10 shows the pairwise interaction strength produced by our proposed method.

**Real data:** We provide the detected interactions for five real datasets in Figure 11. For the Cal housing dataset, we terminate the UCB algorithm when $k = 20$ interactions stood out; for the other four datasets, we terminate at $k = 50$. The green, orange, and blue points represent the UCB, estimated mean, and LCB respectively. For the drug combination dataset, the results are shown in Figure 12.

## K  EMPIRICAL RESULT FOR SAMPLE EFFICIENCY

Figure 13 shows our proposed framework is sample efficient. The data are from the test suite (Table 4), and noise $\eta$ is injected. Increasing the training sample size, both over-parameterized neural nets and our proposed ParaACE perform better. But ParaACE is less data-hungry since fewer training samples

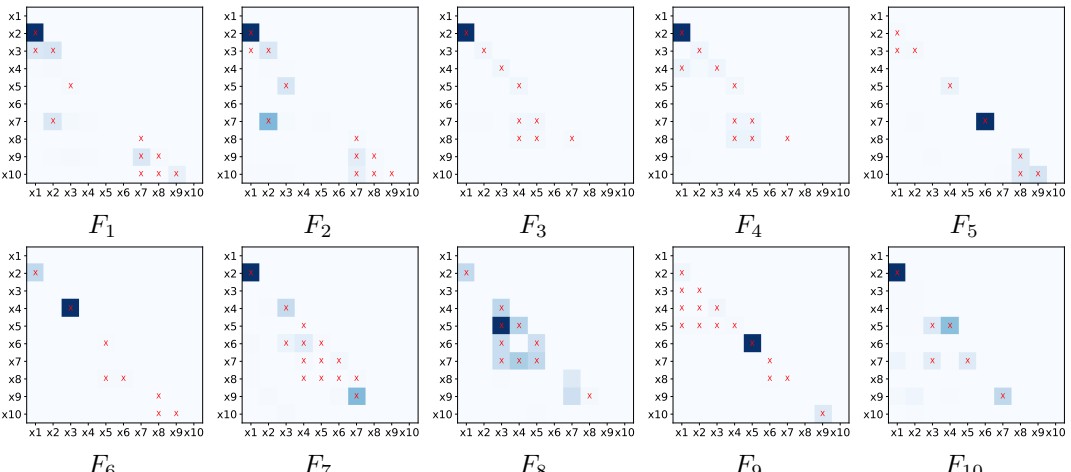

Figure 10: Heat maps of pairwise interaction strength proposed by our method for functions $F_1$-$F_{10}$ (Table 4). Cross-marks indicate the ground truth interactions.

are needed for ParaACE to achieve a similar performance of the over-parameterized neural nets. The experiments show that ParaACE still performs well, even if with small training samples.

## L   DETAILS ON THE COMPARISON WITH KD, LTH, AND SYNFLOW

### L.1   COMPARISON WITH KD

KD is widely used for multi-class classification problems, which extracts knowledge from the "soft label" by controlling the temperature, but fewer KD methods are there for regression problems. Currently, people mainly use the Teacher bounded regression loss (Chen et al., 2017) and the hint loss to deal with the objective detection problem. Passing unlabeled data to the Teacher model to produce pseudo labels can also help. In these ways, the Student is expected to have a similar performance to the Teacher.

We let an over-parameterized FC (10-5000-900-400-100-30-1) be the Teacher, and a ReLU network with architecture (10-70-70-70-70-30-1) be the Student. The Student is trained with 800 original data and 4000 pseudo data (labeled by the Teacher). The loss function we adopted is $L_2$ + Hint loss, where the Hint loss is used on the layer with 30 neurons.

It shows that KD trained with pseudo data and carefully designed loss function achieves similar performance as the Teacher, while ParaACE trained with original data and simple $L_2$ loss achieves significantly better performance.

### L.2   COMPARISON WITH LTH

We used the pruning technique proposed in the lottery tickets hypothesis (LTH) paper (Frankle & Carbin, 2019). We pruned the fully connected neural network after every 20 epochs. Every time 20 % weights were pruned in each layer except for the last layer. We trained 500 epochs, and set the batch size to be 400. We report the best test performance in each round of pruning in Figure 14.

Figure 14 shows that, in most cases pruning the network properly can improve the model performance. But if the network was over-pruned, the performance may drop.

### L.3   COMPARISON WITH SYNFLOW

We implemented the single shot SynFlow for regression tasks based on the official code repository https://github.com/ganguli-lab/Synaptic-Flow (Tanaka et al., 2020). The sparsity is set to $10^{-2.447}$ for synthetic datasets, thus the neural network is compressed 280 times. For real-world datasets, we

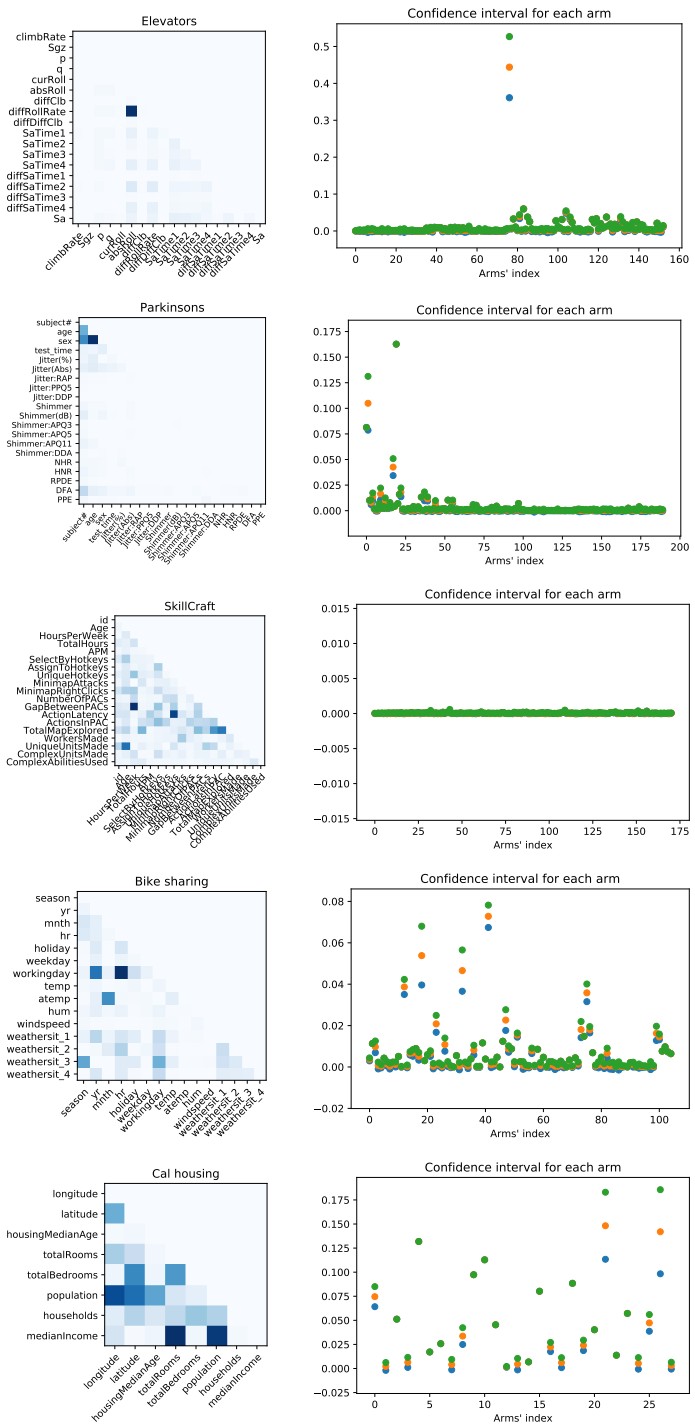

Figure 11: **Left:** Heat maps of pairwise interaction strengths on real datasets. **Right:** Confidence interval for each interaction pair. The green, orange, blue points denote the UCB, estimated mean, and LCB respectively.

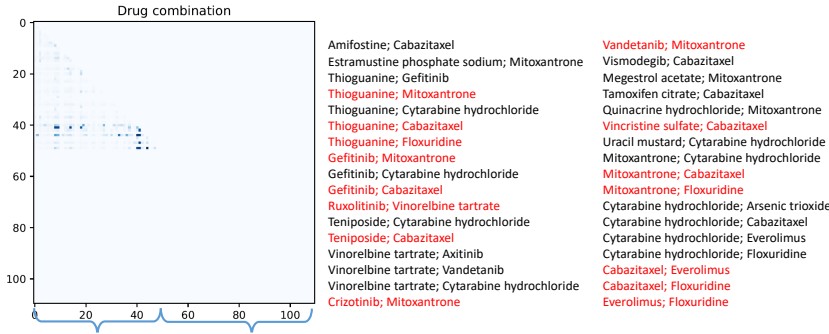

Figure 12: Heat maps of pairwise interaction strengths for drug combination data. The top 34 detected interactions are listed, among which 15 of them (marked in red) are verified in the DrugBank (Wishart et al., 2018).

compressed the model by 120 times. And we retrained the neural network after the single-shot prune (post-training). Note that we did not prune the biases here.

We set the hyper parameters for the post-training process as follows. The batch size is set to be 500, and the number of epochs is 100. We use Adam as the optimizer with lr $= 0.001$, betas $= (0.9, 0.99)$.

## M  ACCURATE APPROXIMATION FUNCTION BENEFITS INTERACTION DETECTION

Suppose the underlying ground truth function is $f(\mathbf{x})$, and our approximation function is $g_{\boldsymbol{\theta}}(\mathbf{x})$, we omit the $\boldsymbol{\theta}$ in the following. We show that the estimation error for Hessian $\left| E_{\mathbf{x}} \left[ \frac{\partial^2 g(\mathbf{x})}{\partial x_i \partial x_j} \right] - E_{\mathbf{x}} \left[ \frac{\partial^2 f(\mathbf{x})}{\partial x_i \partial x_j} \right] \right|$ is bounded by $o(\epsilon)$, when $|f(\mathbf{x}) - g(\mathbf{x})| < \epsilon$ for all $\mathbf{x}$. This implies more accurate pre-trained model leads to better interaction detection accuracy. According to the following theorem, deep neural networks are good surrogate functions due to the universal function approximation capability (Hornik, 1991).

**Theorem M.1.** *Assume $x_i$ are uniformly drawn from $[-1, 1]$ independently, if there exist an $\epsilon > 0$, such that $|f(\mathbf{x}) - g(\mathbf{x})| \le \epsilon$ for all $\mathbf{x} \in \mathbb{R}^p$, then*

$$E_{\mathbf{x}} \left[ \frac{\partial^2 g(\mathbf{x})}{\partial x_i \partial x_j} \right] - E_{\mathbf{x}} \left[ \frac{\partial^2 f(\mathbf{x})}{\partial x_i \partial x_j} \right] \le \epsilon.$$

*Proof.*

$$E_{\mathbf{x}} \left[ \frac{\partial^2 g(\mathbf{x})}{\partial x_i \partial x_j} \right]$$

$$= \int \frac{\partial^2 g(\mathbf{x})}{\partial x_i \partial x_j} p(\mathbf{x}) d\mathbf{x}$$

$$= \int \frac{\partial^2 g(\mathbf{x})}{\partial x_i \partial x_j} p(x_i, x_j) p(\mathbf{x}_{\backslash ij}) d\mathbf{x}$$

$$= \int p(\mathbf{x}_{\backslash ij}) \int_{-1}^{1} \int_{-1}^{1} \frac{\partial^2 g(\mathbf{x})}{\partial x_i \partial x_j} p(x_i, x_j) dx_i dx_j d\mathbf{x}_{\backslash ij}$$

$$\quad [\text{where } p(x_i, x_j) = 1/4]$$

$$= \frac{1}{4} \int p(\mathbf{x}_{\backslash ij}) \int_{-1}^{1} \int_{-1}^{1} \frac{\partial^2 g(\mathbf{x})}{\partial x_i \partial x_j} dx_i dx_j d\mathbf{x}_{\backslash ij},$$

where

$$
\int_{-1}^{1} \int_{-1}^{1} \frac{\partial^2 g(\mathbf{x})}{\partial x_i \partial x_j} dx_i dx_j
$$
$$
= \int_{-1}^{1} \frac{\partial g(\mathbf{x}_{\backslash ij}, x_i = 1, x_j) - \partial g(\mathbf{x}_{\backslash ij}, x_i = -1, x_j)}{\partial x_j} dx_j
$$
$$
= \ g(\mathbf{x}_{\backslash ij}, x_i = 1, x_j = 1) - g(\mathbf{x}_{\backslash ij}, x_i = 1, x_j = -1)
$$
$$
- \ (g(\mathbf{x}_{\backslash ij}, x_i = -1, x_j = 1) - g(\mathbf{x}_{\backslash ij}, x_i = -1, x_j = -1)).
$$

We can derive $E_{\mathbf{x}} \left[ \frac{\partial^2 f(\mathbf{x})}{\partial x_i \partial x_j} \right]$ in a similar fashion. Thus,

$$
E_{\mathbf{x}} \left[ \frac{\partial^2 g(\mathbf{x})}{\partial x_i \partial x_j} \right] - E_{\mathbf{x}} \left[ \frac{\partial^2 f(\mathbf{x})}{\partial x_i \partial x_j} \right]
$$
$$
= \frac{1}{4} E_{\mathbf{x}_{\backslash ij}} [g(\mathbf{x}_{\backslash ij}, x_i = 1, x_j = 1) - f(\mathbf{x}_{\backslash ij}, x_i = 1, x_j = 1)
$$
$$
- \ g(\mathbf{x}_{\backslash ij}, x_i = -1, x_j = 1) + f(\mathbf{x}_{\backslash ij}, x_i = -1, x_j = 1)
$$
$$
- \ g(\mathbf{x}_{\backslash ij}, x_i = 1, x_j = -1) + f(\mathbf{x}_{\backslash ij}, x_i = 1, x_j = -1)
$$
$$
+ \ g(\mathbf{x}_{\backslash ij}, x_i = -1, x_j = -1) - f(\mathbf{x}_{\backslash ij}, x_i = -1, x_j = -1)]
$$
$$
\leq \frac{1}{4} E_{\mathbf{x}_{\backslash ij}} [4\epsilon]
$$
$$
= \epsilon.
$$

$\square$

**Theorem M.2.** *Assume $x_i$ are uniformly drawn from $[-1, 1]$ independently, and further assume $\left| \frac{\partial^2 f(\mathbf{x})}{\partial x_i \partial x_j} \right|, \left| \frac{\partial^2 g(\mathbf{x})}{\partial x_i \partial x_j} \right| \leq M, \forall (x_i, x_j)$. If there exist an $\epsilon > 0$, such that $|f(\mathbf{x}) - g(\mathbf{x})| \leq \epsilon$ for all $\mathbf{x} \in \mathbb{R}^p$, then*

$$
E_{\mathbf{x}} \left[ \left| \frac{\partial^2 g(\mathbf{x})}{\partial x_i \partial x_j} \right|^2 \right] - E_{\mathbf{x}} \left[ \left| \frac{\partial^2 f(\mathbf{x})}{\partial x_i \partial x_j} \right|^2 \right] \leq 2M\epsilon.
$$

*Proof.*

$$
E_{\mathbf{x}} \left[ \left| \frac{\partial^2 g(\mathbf{x})}{\partial x_i \partial x_j} \right|^2 \right] - E_{\mathbf{x}} \left[ \left| \frac{\partial^2 f(\mathbf{x})}{\partial x_i \partial x_j} \right|^2 \right]
$$
$$
= E_{\mathbf{x}} \left[ \left| \frac{\partial^2 g(\mathbf{x})}{\partial x_i \partial x_j} \right|^2 - \left| \frac{\partial^2 f(\mathbf{x})}{\partial x_i \partial x_j} \right|^2 \right]
$$
$$
= E_{\mathbf{x}} \left[ \left( \frac{\partial^2 g(\mathbf{x})}{\partial x_i \partial x_j} - \frac{\partial^2 f(\mathbf{x})}{\partial x_i \partial x_j} \right) \left( \frac{\partial^2 g(\mathbf{x})}{\partial x_i \partial x_j} + \frac{\partial^2 f(\mathbf{x})}{\partial x_i \partial x_j} \right) \right]
$$
$$
\leq \left| E_{\mathbf{x}} \left[ \left( \frac{\partial^2 g(\mathbf{x})}{\partial x_i \partial x_j} - \frac{\partial^2 f(\mathbf{x})}{\partial x_i \partial x_j} \right) \right] \right| \cdot \left| E_{\mathbf{x}} \left[ \left( \frac{\partial^2 g(\mathbf{x})}{\partial x_i \partial x_j} + \frac{\partial^2 f(\mathbf{x})}{\partial x_i \partial x_j} \right) \right] \right|
$$
$$
\leq \epsilon \cdot 2M \quad \text{(Using Theorem M.1)}
$$

$\square$

Note that $M$ can be interpreted as the magnitude of the strongest local interaction. For the functions with strong interactions, higher approximation quality is desired.

Table 9: Ablation study to show the effectiveness of interaction detection on the synthetic datasets in (Table 4). (The results were averaged over 5 folds.)

| | $F_1$ | $F_2$ | $F_3$ | $F_4$ | $F_5$ | $F_6$ | $F_7$ | $F_8$ | $F_9$ | $F_{10}$ | average | CR |
|---|---|---|---|---|---|---|---|---|---|---|---|---|
| ParaACE (with random interaction) | 0.026 | 0.042 | 0.035 | 0.034 | 0.060 | 0.032 | 0.017 | 0.025 | 0.044 | 0.031 | **0.035** | **283** |
| ParaACE (with detected interaction) | 0.025 | 0.035 | 0.031 | 0.030 | 0.038 | 0.025 | 0.016 | 0.019 | 0.023 | 0.021 | **0.026** | **283** |

# N   ABLATION STUDY FOR THE EFFECTIVENESS OF INTERACTION DETECTION

To show how much performance gain we are able to obtain from the interaction detection procedure. We conduct an experiment that input random pairwise interactions to ParaACE to compare with the one with detected interactions on synthetic datasets, see Table 9.

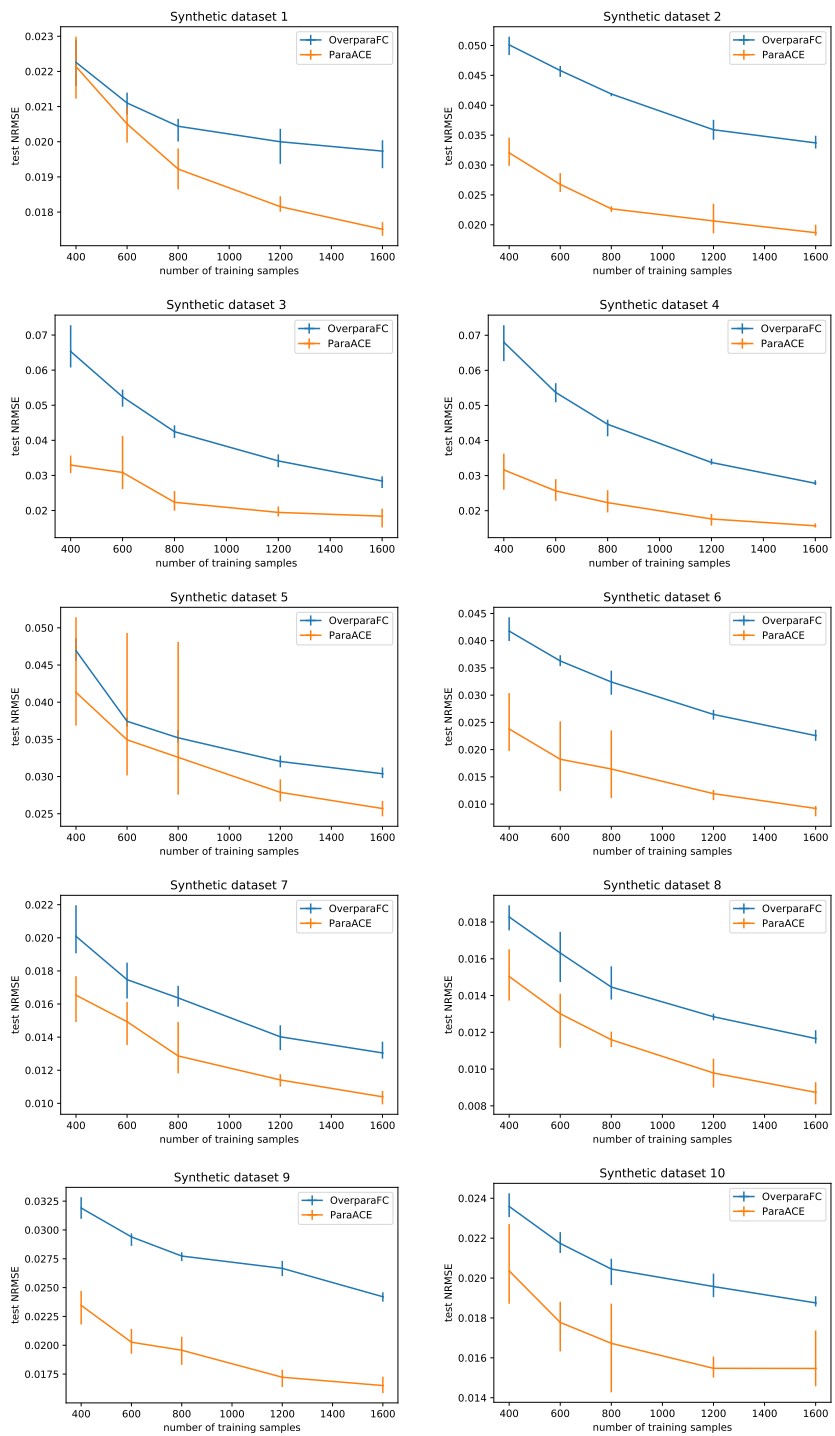

Figure 13: Performance comparison on synthetic datasets while reducing the number of training samples. For each dataset, we tried different training data 5 times in one experiment, and the performances (of ten experiments) were all tested on the same test set. The error bar shows the maximal and minimal test NRMSE in 5 folds.

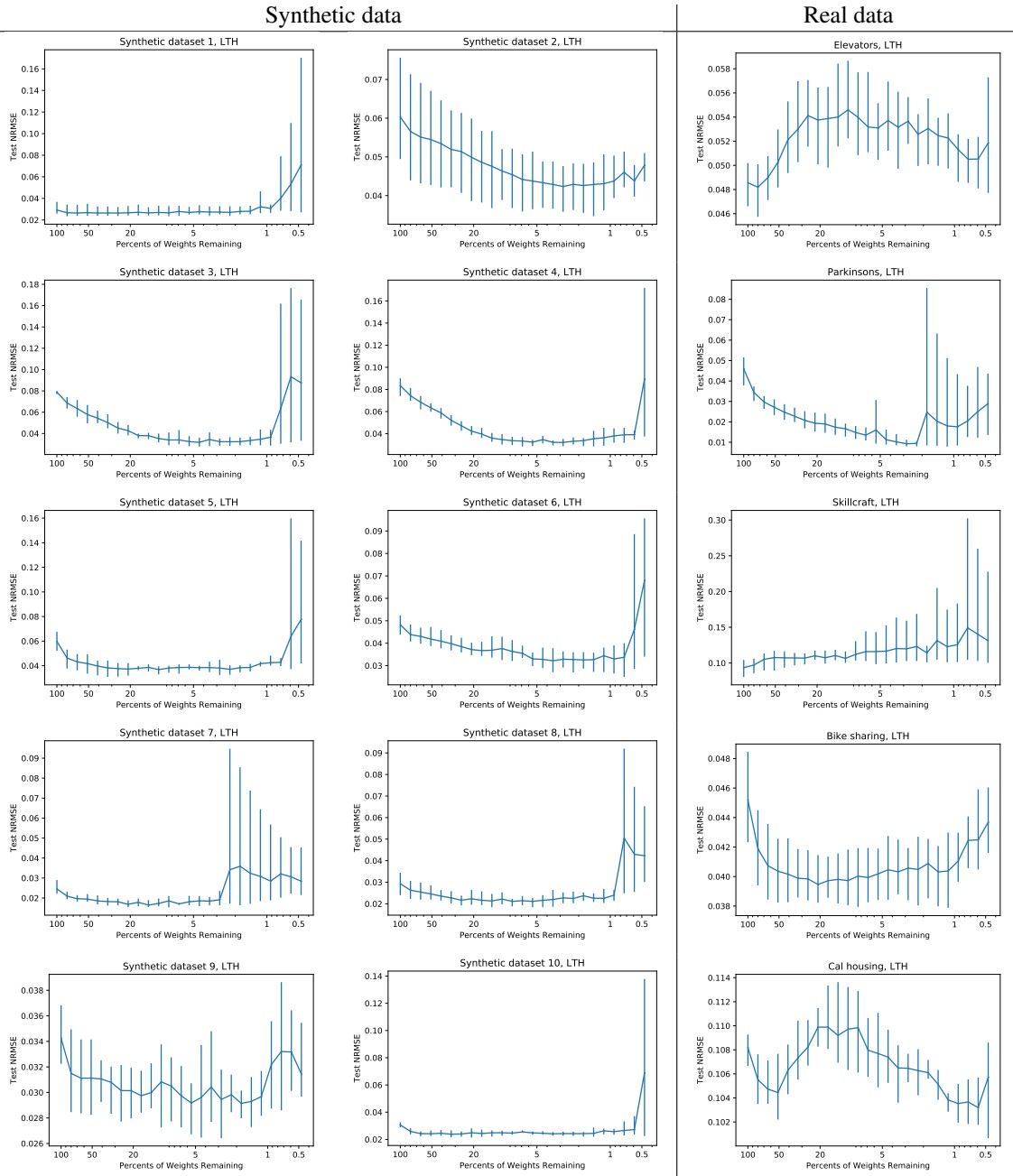

Figure 14: Performance comparison on both the synthetic and real datasets while reducing the percent of weights retained by the LTH. The error bar shows the maximal and minimal test NRMSE in 5 folds.

