# OpenReview forum: "Fast Generic Interaction Detection for Model Interpretability and Compression"
_ICLR.cc/2022/Conference — ICLR 2022 Poster_

### Official Review · Reviewer_QLX7 · 2021-11-02

**Correctness:** 3
**Technical Novelty And Significance:** 4
**Empirical Novelty And Significance:** 4
**Recommendation:** 6
**Confidence:** 4

**Main Review:**

Pros:
+ a fast and principled interaction detection method,
+ a lightweight and interpretable neural network model that can surpass its Teacher,
+ good theoretical analysis and performance evaluations with real datasets.
+ method directly derived from definition of feature interaction
+ the work has taken theory and material from multiple domains and plugged them at apt joints
+ transferring the problem of feature interaction to multi-armed bandits
+ supplement has proofs of statements and explainations
+ apart from synthetic datasets, the real world datasets belong to non-intersecting domains
+ code zip

Cons:
- Upper Confidence Bound (UCB) algorithm is dated 1985 - has nothing else come up interim?
- similarly comparison with classic alternating conditional expectation (ACE) model (1985) feels a bit outdated
- some text book and paper definitions are repeated, instead could have just cited the prior art
- the number of training samples in synthetic data could be increased and tried out at variant degrees
- conclusion section is weak

Miscellaneous:
- theoretical analyses -> theoretical analysis
- F (x), for instance, rewrite
- explain why - an ‘1-regularized ReLU network is required by the latest NID and PID
- The work uses established methods - extract the interaction knowledge by a post-hoc method, and then build a transparent and interpretable learning model - is this combination novel?
- How is interaction different from functional dependency?
- Finite m evaluations of one arm are sufficient to obtain an accurate reward. - is this assumption valid? upper bound of m in practice?
- intelligent, unlike - > interpretable
- parallelizable -> how powerful the smartphone is to support the feature?
- explain more on the cross domain practical applications
- recommended to compare with other state of art approaches in real analysis
- using Matlab logo as function was cool :-)

**Summary Of The Paper:**

The paper handles the important area of model that can be interpreted using feature interactions - aligned to theme of explainable deep learning. The authors chose to use the problem of multi-arm bandit, solving it by UCB algorithm with good speed and accuracy. A lightweight and interpretable deep learning model (called ParaACE), built using alternating conditional expectation (ACE) method is the crux of the work. It is shown that the proposed method improves accuracy by 26% and reduces the model size by 100+ times as compared to its Teacher model over various datasets. The paper has extensive supplementary material, as well as shared code.

**Summary Of The Review:**

This paper is recommended for acceptance at ICLR. The paper is thoroughly written with sound technical backing - especially the code sharing and the content will help in improving the state of art in recent advances in explainable deep learning approaches for others to build on. The applicability in practical use needs more justification.

---

> ### Author Response · Authors · 2021-11-19
> **Feedback to Reviewer QLX7**
>
> Cons：
> - Upper Confidence Bound (UCB) algorithm is dated 1985 - has nothing else come up interim?
>      - In fact, there has been a large number of follow-up works since its publication in 1985. For instance, extensions along various lines, including among others, different UCB formulations and regret analysis subject to additional award constraints and variable costs and/or prior knowledge (Agrawal, 1988; Tran-Thanh, 2010; Ding, 2013, Gupta, 2020; etc.), distributed/collaborative implementations (Liu, 2010; Hille, 2013; Shi, 2021; etc.), Bayesian implementation (Scott, 2010; Kaufmann, 2012), pervasive applications in reinforcement learning in different sectors (too many to enumerate), and various ad-hoc versions based specific assumptions and requirements. The above summary may be biased according to the authors' knowledge and is by no means complete.
>      - In our work, namely for interaction detection, the problem's nature does not suggest incorporating additional assumptions/constraints/prior knowledge. Meanwhile, we hope to present a vanilla version that people can quickly revise for their applications. Therefore, we stick to the original work.
> - Similarly comparison with classic ACE model (1985) feels a bit outdated.
>      - We agree that this ACE work was published a long time ago, but its conveyed idea is not outdated. By finding an optimal set of transformations, the transformed output variable can be expressed as a linearly additive sum of transformed features. In this work, we add interacting terms to enhance its modeling capacity while keeping its interpretability to a certain extent. The simulation results show that even the original version (Breiman et al. 1985) can achieve good data fitting performance. More importantly, we can use the obtained optimal transformation functions to interpret the rationale behind the model, which is missing for most deep learning models. More explanations on the ACE work are given in Supplement H, which might be helpful.
> - Some textbook and paper definitions are repeated, instead could have just cited the prior art.
>      - Thanks for your suggestion. If we were right, you meant Definition 3.1 on page 3. It gives a generic definition of pairwise interaction and serves as the cornerstone of later development. We have made it as short as possible and would instead leave it there for people who are not very familiar with the background of this work.
> - The number of training samples in synthetic data could be increased and tried out at variant degrees.
>      - Thanks for your suggestion. Actually, we showed the effects of increasing training samples in Figure 13. We also have generated new datasets with high-dimensional inputs (50 and 100 features, i.e., p=50 and 100) and a larger training sample size (n=500,000 samples). based on which, we conducted comparisons between our method and the NID method. Results can be found in Supplement F.
> - conclusion section is weak
>      - We have rewritten the conclusion section.
>
> Miscellaneous:
> - theoretical analyses -> theoretical analysis
>   - done!
> - Why the L1-regularized ReLU network is required by the latest NID and PID?
>      - The authors of these two works did not explain the reasons in their papers. But according to our experience, adding L1-regularization can significantly improve the data fitting performance as well as the "accuracy" of their interaction strength measure.
> - The work uses established methods - extract the interaction knowledge by a post-hoc method, and then build a transparent and interpretable learning model - is this combination novel?
>      - We do think the contributions listed at the end of Section 1 are novel.
> - How is interaction different from functional dependency?
>      - They have the same meaning in our work.
> - Finite m evaluations of one arm are sufficient to obtain an accurate reward. - is this assumption valid? upper bound of m in practice?
>      - This assumption essentially implies a sufficient number of training samples, with which the ensemble mean can reasonably approximate the true expectation. To us, this is not a strong assumption and can be easily satisfied in practice.
> - intelligent - > interpretable
>      - done!
> - parallelizable -> how powerful the smartphone is to support the feature?
>      - We have deleted the discussions about parallel implementation as it sounds unrelated to the central theme. But the modern smartphones should train modest-scale deep neural networks that are already good enough for our method.
> - Explain more on the cross-domain practical applications
>      - We have shown some real applications both in the main texts and the supplement. We are not certain what cross-domain applications do you refer to?
> - Compare with other state of the art approaches in real analysis
>      - As also suggested by reviewer akne, we added another pruning method for comparisons. See new results in Tables 2 and 3. A short summary of this method is also given in supplement L.

---

### Official Review · Reviewer_vh8A · 2021-11-03

**Correctness:** 3
**Technical Novelty And Significance:** 3
**Empirical Novelty And Significance:** 2
**Recommendation:** 6
**Confidence:** 3

**Details Of Ethics Concerns:**

minor concern about the usage of the MATLAB logo

**Main Review:**

The desire to consider hessian (and second derivatives in general) as the interaction strength is natural and has been discussed in the literature.
The authors add to this idea more effective estimation by evaluating only those elements of the Hessian matrix that can be valuable.
However, the paper doesn't focus on the effectiveness of this multi-armed approach. Instead, they compare to other approaches for interaction detection from the literature.
Writing is mostly clear, while in some parts it is hard to understand the exact approach, so it can't be reproduced without source code provided.

Pros
* The paper contains a lot of material to explore with many experiments in the main text and in the supplementary
* Theoretical justification on why should it work in a form of a couple of natural theorems.

Cons

Experimental results:
* No experimental for the efficiency of the proposed approach on how does it work compared with the estimation of the full Hessian matrix
* Only a low dimensional case is considered, while the method is designed to outperform others in terms of efficiency in high dimensions. Please, focus on providing evidence, that your method works well in high dimensions.
* It would be interesting to consider simple architectures as an alternative to ParaACE, as other architectures seem to be too complex to learn well for a dataset of size 800. Also, the results will depend on correct applications of regularization techniques. Have you tried them?

Presentation:
* Language, in general, is either too simple in terms of used constructions or too hard to understand. Better to have another round of proofreading. Some examples:
** With the increase of training samples -> Increasing the number of training samples (or "the training sample size")
* No reference to Table 1 in the main text as well as some other tables. Please, provide a reference to them in the main text.
* ROC AUC is a better term for provided scores than AUC, as you consider the ROC curve for the interaction detection
* Figure 13: you have no uncertainty in sample size, so it should be not crosses, but vertical uncertainty bars (by the way, what are they? One, two, three STDs?)
* I suppose, that you don't own the rights for the MATLAB logo (Figure 6), so maybe you should cite the figure source?
* Figure 5: what is the color bar? Does darker blue color correspond to stronger interaction?

Theoretical results:
Theorem M.1 says nothing about absolute values of the elements of the Hessian matrix, only with non-absolute values.
So, if the approximation quality is good, in reality we can have pretty bad estimate for an integral $E_x |\frac{\partial^2 F(x_1, x_2)}{\partial x_1 \partial x_2}|$. For example, you can consider $F(x_1, x_2) = \sin(w(x_1 + x_2))$. While this function lies in $[-1, +1]$, the intergral is proportional to $w^2$, which can take arbitrary large values.

Hard to reproduce:
* Distillation approach never defined. As there are plenty of them out there, please specify, which one you have used. What is the exact loss function?
* The notation in the definition of the NN architecture is not clear: p-5000-900-400-100-30-1, 1-50-8-1, 2-50-8-1 don't explicitly define the architecture
* What is a ReLU network - unclear
* Single layer ResNet - not defined

Design choices:
* Why Kaiming’s strategy for initialization?
* Typical batch size is a power of 2. Why 500 in your case?

**Summary Of The Paper:**

The paper proposes to detect pairwise estimation using a more efficient evaluation of Hessian values via sampling based on the multi-armed bandit approach.

**Summary Of The Review:**

The authors propose a more efficient method for hessian matrix estimation with little novelty and not comprehensive experimental results.
If experimental issues are resolved with experiments on real datasets of high dimension and comparison with a proper baseline, then there is a change for the acceptance.

---

> ### Author Response · Authors · 2021-11-19
> **Feedback to Reviewer vh8A**
>
> Thank you so much for your careful reading and constructive review comments. We have carefully addressed all your questions below and revised the manuscript accordingly. We sincerely hope that our answers fully address your concerns.
> - For the suggestions on our experimental results, we clarify as follows.
>      1.  The results of efficiency are shown in Figure 8. Pulling each arm more than 100 times (The total number of pulls is thus 100*45=4500) can achieve accurate enough detection results (represented by the bright yellow bricks). However, our proposed algorithm only needs around 1500 pulls.
>      2.  For the high-dimensional case, we generated new datasets with 50-dimensional and 100-dimensional input features, respectively (reported in Supplement F). In contrast, UCB pulls vs naïve pulls is around 21,000 vs 122,500 for 50-dimensional dataset and 40,000 vs 495,000 for 100-dimensional dataset.
>
>      3. Consider simple architectures as an alternative to ParaACE. Also, the results will depend on the correct applications of regularization techniques. Have you tried them?
>
>         -  Yes, we have ever considered the simple linear model and generalized additive model (GAM)(using the R gam package) with expanded pairwise features $x_ix_j$. But both of them perform worse than our proposed ParaACE model.
>
>           |                         | F1    | F2    | F3    | F4    | F5    | F6    | F7    | F8    | F9    | F10   | average |
>           |-------------------------|-------|-------|-------|-------|-------|-------|-------|-------|-------|-------|---------|
>           | Lasso+expanded features | 0.036 | 0.102 | 0.108 | 0.115 | 0.095 | 0.111 | 0.031 | 0.036 | 0.053 | 0.039 | 0.073   |
>           | GAM+expanded features   | 0.048 | 0.081 | 0.069 | 0.067 | 0.094 | 0.077 | 0.036 | 0.051 | 0.054 | 0.049 | 0.063   |
>
>         -  We didn't use any explicit regularizations for all the neural networks mentioned in the paper for fair comparison; otherwise, it is difficult to tell the performance gain is from the **ParaACE architecture** or **the regularization tricks**. We think the stochastic gradient-based optimization can be regarded as an implicit regularization. We note that for ParaACE, we did leave an option to regularize the $\boldsymbol{\beta}$ vector (the last layer in ParaACE) in the code for a further selection of the interaction pairs.
>
> - For the presentation part, we improved the statements as you suggested in the updated version (language, reference, ROC-AUC, error bar).
>      - Short answers to your questions:
>           1. The error bar shows the maximal and minimal test NRMSE in 5 folds.
>           2. We used the L-shaped membrane symbol with permission from MathWorks.
>           3. The darker blue color in the heatmap corresponds to stronger interaction.
>
>
> -	Comment on the example, F(x1,x2)=sin(w(x1+x2)).
>      -	Thanks for the valuable discussion. We provided Theorem M.2 in the updated draft to address your question. Intuitively, with the frequency $\omega $ increasing, the strength of the interaction is stronger. The upper bound of the difference of $E_x|\frac{\partial^2 f(x_1,x_2)}{\partial x_1 \partial x_2}|^2$ and $E_x|\frac{\partial^2 g(x_1,x_2)}{\partial x_1 \partial x_2}|^2$ scales with $\omega^2$. That means, for stronger interactions (higher frequency in your example), the approximation quality is more important. Otherwise, the difference between $E_x|\frac{\partial^2 f(x_1,x_2)}{\partial x_1 \partial x_2}|^2$ and $E_x|\frac{\partial^2 g(x_1,x_2)}{\partial x_1 \partial x_2}|^2$ will blow up. This supports our comments in the paper, “The goodness of $F(\mathbf{x})$ as an approximator can essentially influence the interaction detection performance”. We hope this explanation is clear.
>
>
>
> - For "Hard to reproduce",
>      - We did include the descriptions of KD in Section L.1 of the initial submission and provided the code in the initial zip file (locating in the folder ./experiments_mxnet).
>
>      - We added a description for the architectures (e.g., p-5000-900-400-100-30-1) in Section 5.1.
>      -	ReLU network means multilayer perceptron with ReLU activation here.
>      -	We define the single layer ResNet as $F(x)=x+W_2ReLU(W_1x+b_1)+b_2$.
>
>
> - For "Design Choices",
>      -	Kaiming's strategy is beneficial to deep ReLU networks, which can alleviate the problem of gradient vanishing/exploding. Of course, one can try out other strategies. In practice, we may try different initial guesses and pick out the best one from cross-validation.
>      -	We do not think there is a strict requirement of choosing the batch size as a power of 2. The number of our training samples is 10000, and it can be divided by 500 exactly. That's why we choose 500 as the batch size.
>
> -	Compare with a proper baseline.
>      -	As suggested by Reviewer akne, we added one more pruning algorithm (Synaptic Flow) as a stronger baseline in both Table 2 and Table 3.

---

> > ### Comment · Reviewer_vh8A · 2021-11-24
> > **Thanks for the update**
> >
> > I like more the new version of the paper with the theorem issue addressed and a new baseline added. I will increase my score, as now I see how the proposed approach can contribute to the general community on interaction detection, being simple, while rather effective.

---

> ### Author Response · Authors · 2021-11-22
> **A quick summary of the response below**
>
> Dear Reviewer vh8A,
>
> Thanks again for the helpful discussion and comments. Here is a quick summary of the response below.
> 1. We fixed all the presentation flaws mentioned by you. No more copyright issues. :-)
> 2. We improved our Theorem M.1 (see the new one in Theorem M.2), as you made an excellent point with the intuitive example of $F(x_1,x_2)=sin(w(x_1+x_2))$.
> 3. We reported the experimental results with synthetic datasets to justify that our proposed UCB-based interaction detection method is effective and efficient even in high-dimensional cases. We didn't have the experiments for real datasets since it is difficult to find a high-dimensional real dataset with fully labeled interaction pairs (ground truth). We hope the results are comprehensive to you.
> 4. We added one more substantial baseline, SynFlow (https://arxiv.org/abs/2006.05467), for pruning in Table 2 (synthetic datasets) and Table 3 (real datasets).
>
> We hope the response is sufficient. Please do not hesitate to discuss with us if you have any doubts.
>
> Sincerely,
>
> The authors

---

### Official Review · Reviewer_bpD2 · 2021-11-05

**Correctness:** 3
**Technical Novelty And Significance:** 3
**Empirical Novelty And Significance:** 4
**Recommendation:** 8
**Confidence:** 4

**Main Review:**

The paper has several strong points as follows:

- the authors propose the algorithm by building up from the first principles i.e. by inspecting the expected Hessian. Furthermore, the proposed finite difference method holds the promise of being a true model agnostic implementation. Such methods are of high importance and can be used in settings where the model is black-box by design or by method of delivery (e,g. a binary file)
- The simplification of the problem to best k arms leads to a more computationally feasible solution. This is a novel contribution and the authors also explored this by conducting several theoretical analysis
- The model compression algorithm presented by the authors is very interesting. By itself, its an interesting contribution and could perhaps be used independent of the interaction detection method. The performance achieved by this method is also highly promising (read more on this below).
- Finally, the analysis on the consistency of the discovered interactions is very well presented and provides an insight into how the method can be used for other settings in a debug and develop method


The paper can be improved upon by addressing some of the aspects below:

- The claim about model agnosticity of the method seems to be well supported. It would be also interesting if the authors can discuss whether their implementation can be framework agnostic e.g. can work with scikit-learn, pytorch, tensorflow models? The proposed methods seems to be gradient free and thus it would be interesting to comment on this
- The assumption in 3.3 b sounds interesting. Can the authors provide more insights into why the draws can be considered coming from independent reward distribution? E.g. if there are higher level interactions present, would this first order interaction be somehow impacted by the order of the draw?
- The presentation of the paper can be improved upon as well. For example, in table 1, the authors can consider annotating the methods by whether these are "post-hoc"/ "model agnostic" / "statistical methods" and so on
- While the model compression method presented is very interesting, the claims about interpretability of the proposed model may need to be reduced and/or substantiated. The fix-up layer seems to introduce non-linear transformations and its not immediately apparent how would the model interactions directly be interpretable in that setting. For example, would a simple GAM on the interactions and their proposed model be equivalent? In general, the claims of interpretability  (valid from the interactions, perhaps unsubstantiated on the ParACE model) and compressibility may need to clearly separated out

On some other aspects, its not apparent why the ParACE model would provide such a boost in performance. Is this an artifact of the selected datasets or would such a boost be expected in general? It would be interesting if the authors can provide some insights into this aspect

**Summary Of The Paper:**

Identification of feature interactions from black-box models, on a global as well as on a local scale, can lead to a better understanding of the operating modes of the black-box models. In this paper, the authors presented a principled approach to identify global interactions by casting the problem as a multi-arm bandit problem and proposes a solution using UCB algorithm. The authors claim that their method interaction discovery method is free of ad-hoc assumptions with good detection accuracy and stability. Furthermore, the authors showcase the importance of the learned interactions by proposing a new deep learning model based on these interactions and showcase the improvements in model size (thereby competing against pruning methods) as well as in accuracy (thereby competing against generalization methods).

**Summary Of The Review:**

Overall this is a very nice paper. Identification of interaction terms can lead to very novel analysis of the black-box models and can even lead to knowledge discovery e.g. discovering drug-drug interactions. The paper also seems to have 2 strong contributions where the proposed ParACE model leads to a very novel compressible architecture. The paper can be improved upon by addressing some of the aspects but even in its current state it will be of interest to the larger AI/ML community.

---

> ### Author Response · Authors · 2021-11-19
> **Feedback to Reviewer bpD2**
>
> Thank you so much for your careful reading and constructive review comments. We have carefully addressed all your questions below and revised the manuscript accordingly. We sincerely hope that our answers fully address your concerns.
>
> - (1) Whether the implementation can be framework agnostic, e.g., can work with Scikit-learn, Pytorch, Tensorflow models?
>
>      - Thanks for the comments. Our current implementation is compatible with Pytorch, Mxnet (for NN model), and Scikit-learn (for RF model). We believe that our implementation can be used for a broader class of models with slight modification.
>
> - (2) Comments on independent draws from the reward distribution.
>      - We wrote: "b. The reward for each arm is drawn independently and identically from its reward distribution". The reward refers to $(\frac{\partial^2F}{\partial x_i\partial x_j})^2$ evaluated on a data sample drawn uniformly from the dataset. The independence comes from the uniform draws.
>
> - (3) We have annotated the properties of different benchmarks by using footnotes in Table 1. We didn't include the "statistical method" since it may confuse the readers.
>
> - (4) While the model compression method presented is very interesting, the claims about interpretability of the proposed model may need to be reduced and/or substantiated.
>      - We agree that the interpretability may lose to a certain extent, due to the nonlinear transformation introduced by the Fix-up layer. By choosing the ResNet, we actually aim to weaken the negative effect of nonlinearity on interpretability. Moreover, this fix-up layer is optional if one targets more on interpretability. The experimental results obtained from synthetic datasets have already demonstrated good interpretability. As pointed by one Reviewer, this work does not overclaim the interpretability aspect of the ParaACE.
>
> -	Why the ParACE model would provide such a boost in performance?
>      - Our choice of test datasets (both synthetic and real) aligns with the literature and covers different application domains. Due to the good nature of our method, we firmly believe that our method can be generalized to most datasets.
> Possible reasons are as follows：
>      1. As shown in the figures, most decompositions of the underlying function are well captured by the optimal transformations found by the ParaACE.
>      2. Negative effects from mis-detected higher-order interactions and wrongly detected pairwise interactions will be reduced by the fix-up layer.
>      3. The whole ParaACE method can be regarded as one novel and very effective neural network pruning method---where there is an effective interaction, there is a link.

---

### Official Review · Reviewer_akne · 2021-11-06

**Correctness:** 3
**Technical Novelty And Significance:** 3
**Empirical Novelty And Significance:** 2
**Recommendation:** 6
**Confidence:** 4

**Main Review:**

Strengths:
- To my knowledge, no one has applied a multi-armed bandits approach to interaction detection
- The ParaACE model is a nice idea for model compression, and it shows promise. The fixup layer, in particular, is a clever way to try to include higher order interactions to recover the performance of the full model. Most papers that use identified interactions to build a new model (e.g. https://www.cs.cornell.edu/~yinlou/papers/lou-kdd13.pdf) are primarily focused on creating interpretable models (not compressed models), and thus those papers do not include anything like a fixup layer (which breaks interpretability by including higher-order interactions). Thus, the very idea of extracting the pairwise interactions for the purpose of creating a **compressed** model (rather than an interpretable model) is, to my knowledge, novel - and I think difference in objectives is worth highlighting more prominently.
- In a similar spirit to the point above, the authors show that ParaACE often surpasses the performance of its overparamerized teacher model; once again, this is a relatively novel angle to models constructed using learned feature interactions, because most such models are concerned about **matching** the teacher performance and rarely concerned about exceed it. By forfeiting interpretability with the fixup layer, ParaACE shows that constraining the model with the ParaACE architecture is a smart way to combat overfitting.
- The authors have done a good amount of technical work to evaluate their approach in different contexts.

Weaknesses:
1. My main concern is that there appears to be a much more computationally efficient baseline that the authors have overlooked. The authors estimate the Hessian using the method of finite differences (equation 1), which requires four model evaluations to get one entry in the Hessian matrix. For neural networks, there exists an efficient way to get the entries for an entire **row** of the Hessian as follows: first, compute the gradient $\partial F(\boldsymbol{x})/\partial x_i$ for all $x_i$ in a standard backpropagation pass. Then, add a small perturbation $h$ to some input $x_j$. After making the perturbation, recompute the gradients $\partial F(\boldsymbol{x} + \boldsymbol{e}_j h)/\partial x_i$ for all $x_i$ in another standard backpropagation pass (as in the text, $\boldsymbol{e}_j$ is a one-hot vector with the $j$th entry set to 1 and zero elsewhere). Estimate the Hessian entries for the row as $\frac{1}{h} [\partial F(\boldsymbol{x} + \boldsymbol{e}_j h)/\partial x_i - \partial F(\boldsymbol{x})/\partial x_i]$. This approach fills in the entire $j$th row of the Hessian and requires only two forward-backward passes through the model. This is the idea adopted in https://academic.oup.com/bioinformatics/article/34/17/i629/5093210, though I suspect references to this approach exist even earlier in the literature. It is true that the multi-armed bandit approach could likely be adapted to using this technique to fill out the hessian matrix (perhaps now the "arms" would correspond to rows in the hessian, rather than individual entries in the hessian?), though I am not sure how dramatic the computational gains would be relative to the naive approach.

2. Elaborating more on my concerns re the computational complexity of the current UCB approach: the authors write that in their experiment with only 10 features, "our proposed method needs around 1500 pulls of arms to pick out the top 20 interactions, however,
naively pulling each arm 100 times needs 4500 pulls in total" - and yet, using the approach I suggested in (1.) with a naive 100 pulls per row of the hessian, the total number of pulls needed would be 1000, which is *smaller* than the 1500 pulls needed for finding only the top 20 interactions. This difference would be even more dramatic for experiments with even more features. The authors mention experiments with more features in supplement F - however, it looks like the authors only discuss the AUCs and *don't mention the number of pulls needed for these larger experiments*. I think it's important to report the number of pulls.

3. Regarding the fact that the Hessians are evaluated on examples randomly drawn from the training data: this could conceal cases where the Hessians are near 0 due to saturation effects (this is analogous to the gradient saturation problem in the context of feature importance scoring method (Figure 1 in the DeepLIFT paper https://arxiv.org/pdf/1704.02685.pdf)). However, I recognize that, depending on the perturbation size h, this saturation issue could be overcome. Still, it is worth considering whether to adopt an approach like the one in https://academic.oup.com/bioinformatics/article/34/17/i629/5093210, where interaction strength was defined as the change in the feature importance of feature i in response to perturbations in j (and "feature importance" could be computed using alternatives to the gradient that are less likely to suffer from saturation issues); even though it is more heuristic, it may bring down the number of arm pulls needed to detect a true feature interaction.

4. In terms of benchmarking the compression offered by ParaACE, I think there are successors to the LTH approach that are worth including in the benchmarks to make them more compelling - e.g. "Pruning neural networks without any data by iteratively conserving synaptic flow" (NeurIPS 2020: https://arxiv.org/abs/2006.05467)

5. A minor but important note regarding the extent to which ParaACE is interpretable: note that a pairwise interaction subnetwork is capable of learning main effects of features too; thus, ParaACE may not be that interpretable in terms of decomposing the prediction into main effects and interaction effects.

Minor
- The authors called ParaACE "intelligent" (bottom of page 6) - I think they meant "intelligible" or "interpretable"?
- The authors mention "smartphones" in the context of the parallelizability of ParaACE - not sure what the connection is; smartphones are typically mentioned for portability rather than parallelizability?


**Summary Of The Paper:**

The authors propose two key ideas. The first is the idea of using the UCB algorithm to identify strong feature interactions in a computationally efficient way; each set of interacting features is an "arm" that could be "pulled", and pulling the arm corresponds to evaluating the strength of the interaction by computing the corresponding entry in the Hessian on a random training example. The finite difference method is used for computing Hessians. The second key idea is that of using the identified pairwise interactions to build a lightweight GAM-like model that they call ParaACE. In experiments, the authors show the UCB approach is effective at identifying feature interactions compared to alternative methods, and they demonstrate that ParaACE offers strong gains in model compression compared to other competing approaches, and often improves performance relative to its overparameterized teacher model.

**Summary Of The Review:**

I think the work proposes several interesting ideas; these ideas may be based on prior literature and may seem straightforward in hindsight, but I think they involved creative thinking to propose in the first place. Although I am not convinced by the computational efficiency of the proposed UCB algorithm and think a more computationally efficient baseline is missing from the comparisons, I think the idea of taking a multi-armed bandits approach to model interpretation is still clever, and the UCB algorithm could likely be extended/modified to leverage a more computationally efficient way of evaluating interactions. I also think the ParaACE approach to compressing models and combatting overfitting is a novel take on how feature interactions can be made useful (prior work has been heavily focused only on interpretability). However, at least for model compression, I think there is at least one baseline method worth including. Overall, I rate the paper above the acceptance threshold as-is.

---

> ### Author Response · Authors · 2021-11-19
> **Feedback to Reviewer akne**
>
> Thank you so much for your careful reading and constructive review comments. We have carefully addressed all your questions below and revised the manuscript accordingly. We sincerely hope that our answers fully address your concerns.
> - (1&3) Is it possible to compute the first-order gradient first (using backpropagation) and then use the perturbation-based method?
>      - Your comments make sense; however, our framework covers a more general case when automatic differentiation (AD) is unavailable (e.g., RF model).
> As summarized in Figure 5 of Baydin et al. (2018), reverse AD is much more efficient than numerical differentiation in high dimensional cases. With this technique, only $2p$ times forward-backward are needed to calculate the full Hessian evaluated at one data point. Indeed, this is an excellent proposal for gradient-based models. We noted this point in the description of numerical evaluation in Section 3.2.
> Still, the multi-arm bandit approach can be adapted to this context.
>
> - (2) "I think it's important to report the number of pulls."
>      - Great suggestion! The number of UCB pulls is model dependent, as reported in our submitted `.ipynb` file (locating in the folder ./experiments_mxnet/NID_vs_Ours_50_100), which shows that the number of UCB pulls vs naïve pulls is around 21,000 vs 122,500  (for 50-dimensional dataset) and 40,000 vs 495,000 (for 100-dimensional dataset). For quicker access, We have added these results in the revised manuscript.
>
> - (4) Use Synaptic Flow as a stronger baseline.
>     - Synaptic Flow (SynFlow) is well-known for overcoming the so-called layer-collapse and reducing the retraining process's cost. We added single-shot SynFlow for comparison and reported fresh results in Table 2 and Table 3. The overall performance of SynFlow is better than LTH but weaker than ours.
>
> - (5) The main effects may be learned in the interaction.
>      - We agree that the main effects might be absorbed into the interaction transformation. It is not clear how to completely avoid it in practice.
>
> - Minor
>      - intelligent -> intelligible/interpretable Done!
>      - "The authors mention "smartphones" in the context of the parallelizability of ParaACE - not sure what the connection is."
>           - We have deleted the discussions about parallel implementation as it sounds unrelated to the central theme. But modern smartphones should train modest-scale deep neural networks that are already good enough for our method.

---

> > ### Comment · Reviewer_akne · 2021-11-22
> > **Thanks for the additions**
> >
> > Thank for adding in SynFlow and reporting the number of pulls. I will likely increase my score - ideally I would have already increased it from a 6 to a 7, but the only option for increasing the score is 6->8 in the current rating system, so I feel I need to go over the paper one more time before I give it an 8 (which to me is a pretty rare score). But take this comment as a signal to the metareviewer that the score has been increased implicitly. And I agree with the response to (1 & 3).
> >
> > "We agree that the main effects might be absorbed into the interaction transformation. It is not clear how to completely avoid it in practice" -> see this paper “Purifying Interaction Effects with the Functional ANOVA: An Efficient Algorithm for Recovering Identifiable Additive Models” https://arxiv.org/abs/1911.04974
> >
> > Cheers
> > - reviewer akne

---

> > > ### Author Response · Authors · 2021-11-23
> > > **On interaction effect purification**
> > >
> > > Thanks for the suggestion on purifying the interaction effect. The paper defines the **pure interaction effect** as the variance in the outcome that fewer variables cannot describe. And the tree-based models are purified by calculating fANOVA. We do think this is a valuable point and can be adopted to enhance our ParaACE model in future work. Possibly we can estimate a piecewise-constant function $\hat{F}$ for ParaACE first, and then apply the purifying algorithm directly. One huge advantage is that **the number of bins** required for the piecewise-constant approximation $\hat{F}$ can be **reduced significantly**, since we have access to univariate and bi-variate transformations.   We will mention this valuable idea and add the associated reference in the revised manuscript.

---

### Decision · Program_Chairs · 2022-01-20

**Decision:**

Accept (Poster)

**Comment:**

This paper tackles the problem of feature interactions identification in black-box models, which is an important problem towards achieving explainable AI/ML. The authors formulate the problem under the multi-armed bandit setting and propose a solution based on the UCB algorithm. This simplification of the problem leads to a computationally feasible solution, for which the authors provide several theoretical analyses. The importance of the learned interactions is showcased in a new deep learning model leveraging these interactions, leading to a reduction in model size (thereby competing against pruning methods) as well as an improvement in accuracy (thereby competing against generalization methods). Although the proposed approach essentially builds on the specific UCB algorithm, it could likely be extended/modified to other (potentially more efficient) bandit strategies. A drawback of this work resides in the experiments being entirely synthetics. In order to close the gap with practice, experiments on real datasets of higher dimensionality should be conducted.